# Positive Effects of Exercise Intervention without Weight Loss and Dietary Changes in NAFLD-Related Clinical Parameters: A Systematic Review and Meta-Analysis

**DOI:** 10.3390/nu13093135

**Published:** 2021-09-08

**Authors:** Ambrin Farizah Babu, Susanne Csader, Johnson Lok, Carlos Gómez-Gallego, Kati Hanhineva, Hani El-Nezami, Ursula Schwab

**Affiliations:** 1School of Medicine, Institute of Public Health and Clinical Nutrition, University of Eastern Finland, 70200 Kuopio, Finland; ambrin.babu@uef.fi (A.F.B.); susanne.csader@uef.fi (S.C.); johnson.lok@uef.fi (J.L.); carlos.gomezgallego@uef.fi (C.G.-G.); kati.hanhineva@uef.fi (K.H.); hani.el-nezami@uef.fi (H.E.-N.); 2Afekta Technologies Ltd., Yliopistonranta 1L, 70211 Kuopio, Finland; 3Department of Life Technologies, Food Chemistry and Food Development Unit, University of Turku, 20500 Turku, Finland; 4School of Biological Sciences, The University of Hong Kong, Pokfulam Road, Hong Kong 999077, China; 5Department of Medicine, Endocrinology and Clinical Nutrition, Kuopio University Hospital, 70210 Kuopio, Finland

**Keywords:** systematic review, meta-analysis, non-alcoholic fatty liver disease, aerobic exercise, resistance training, lipid metabolism, glucose metabolism, transaminases

## Abstract

One of the focuses of non-alcoholic fatty liver disease (NAFLD) treatment is exercise. Randomized controlled trials investigating the effects of exercise without dietary changes on NAFLD-related clinical parameters (liver parameters, lipid metabolism, glucose metabolism, gut microbiota, and metabolites) were screened using the PubMed, Scopus, Web of Science, and Cochrane databases on 13 February 2020. Meta-analyses were performed on 10 studies with 316 individuals who had NAFLD across three exercise regimens: aerobic exercise, resistance training, and a combination of both. No studies investigating the role of gut microbiota and exercise in NAFLD were found. A quality assessment via the (RoB)2 tool was conducted and potential publication bias, statistical outliers, and influential cases were identified. Overall, exercise without significant weight loss significantly reduced the intrahepatic lipid (IHL) content (SMD: −0.76, 95% CI: −1.04, −0.48) and concentrations of alanine aminotransaminase (ALT) (SMD: −0.52, 95% CI: −0.90, −0.14), aspartate aminotransaminase (AST) (SMD: −0.68, 95% CI: −1.21, −0.15), low-density lipoprotein cholesterol (SMD: −0.34, 95% CI: −0.66, −0.02), and triglycerides (TG) (SMD: −0.59, 95% CI: −1.16, −0.02). The concentrations of high-density lipoprotein cholesterol, total cholesterol (TC), fasting glucose, fasting insulin, and glycated hemoglobin were non-significantly altered. Aerobic exercise alone significantly reduced IHL, ALT, and AST; resistance training alone significantly reduced TC and TG; a combination of both exercise types significantly reduced IHL. To conclude, exercise overall likely had a beneficial effect on alleviating NAFLD without significant weight loss. The study was registered at PROSPERO: CRD42020221168 and funded by the European Union’s Horizon 2020 research and innovation program under the Marie Skłodowska-Curie grant agreement no. 813781.

## 1. Introduction

Non-alcoholic fatty liver disease (NAFLD) is the most common chronic liver disease, affecting around 25% of the world’s population [1] and 23% of the European population [2]. It encompasses liver conditions ranging from benign steatosis to non-alcoholic steatohepatitis (NASH), including inflammation with or without fibrosis [3]. Moreover, NASH can progress to liver cirrhosis and liver cancer [4,5]. The occurrence of NAFLD is rising due to unhealthy lifestyles, including sedentary lifestyles and increased energy intake [6]. In addition, the gut microbiota and the metabolites they produce are also known to contribute toward NAFLD development and progression [7,8].

Despite the increasing occurrence of the disease, no approved pharmacological therapies for NAFLD treatment exist so far. Hence, the first step to improving NAFLD pathology in its early stages is focused on lifestyle modifications [9]. These modifications include increased physical activity and consumption of a healthy diet with restricted intake of saturated fat and controlling body weight and cardiometabolic disorders related to metabolic syndrome [9,10]. For example, a combination of hypocaloric diet and exercise resulted in significant improvements in anthropometric indices, total cholesterol, insulin sensitivity, liver biochemistry, ultrasonography (US) findings of the liver, and physical fitness [11].

It is not only combined diet and exercise intervention studies that have shown improvements in NAFLD development and progression but also those focusing on exercise alone [12,13]. Exercise alone (with or without weight loss) was shown to improve the clinical parameters of NAFLD, such as intrahepatic lipid (IHL) content, insulin sensitivity in skeletal muscle [14,15,16], and liver enzymes, such as alanine aminotransaminase (ALT) and asparagine aminotransferase (AST) [17,18]. However, the results of exercise studies remain inconsistent [19]. Aerobic exercise alone was shown to improve body weight, glycated hemoglobin (HbA1c), blood pressure, and cholesterol concentration [20]. These aerobic exercises range from low (walking) to high intensity (running, swimming) and focus on cardiovascular conditioning. On the other hand, resistance training exercises focus on increasing muscle strength by working the muscles against a force. Such exercises were shown to decrease blood pressure and IHL and improve insulin resistance [19,21]. Moreover, a combination of aerobic exercise and resistance training was also shown to improve the metabolic parameters known to interfere in NAFLD development and offer a protective role against it [22].

The European Association for the Study of the Liver (EASL) recommends physical activity of 150–200 min per week to improve metabolic health [9]. However, these recommendations do not specify what exercise regimen is the most beneficial for NAFLD. Therefore, this meta-analysis aimed to evaluate the contribution of different exercise regimens for different clinical parameters (IHL content, ALT, AST, high-density lipoprotein cholesterol (HDL-C), low-density lipoprotein cholesterol (LDL-C), total cholesterol (TC), triglyceride (TG), fasting glucose, fasting insulin, and HbA1c). Furthermore, exercise also influences the gut microbiome and its metabolites, and their potential contribution to improvements in metabolic disorders was proposed [23]. Therefore, the relationship between exercise-induced changes in the gut microbiome and metabolite composition and NAFLD was also considered in our study. Moreover, several studies showed that in order to alleviate NAFLD, a certain level of weight reduction is necessary [24,25]. To the best of our knowledge, this systematic review (SR) and meta-analysis are the first to demonstrate the positive effects in NAFLD subjects upon exercise intervention without significant weight loss and changes to their diet.

## 2. Materials and Methods

### 2.1. Protocol and Registration

The results of this SR are presented according to the Patient, Intervention, Comparison, and Outcome statement (PICO) and to the Preferred Reporting Item for Systematic Review and Meta-Analysis (PRISMA) (PRISMA checklist Appendix A) [26]. The review protocol was registered in Prospero (CRD42020221168).

### 2.2. Data Sources and Search Strategy

A systematic literature search was performed to gather all randomized clinical trials (RCTs) that analyzed the effects of exercise (and other physical activities) in patients that were clinically diagnosed with NAFLD. A comprehensive search of the PubMed, Scopus, Web of Science, and Cochrane databases was first performed from the inception of each database until February 13, 2020, using keywords such as “non-alcoholic fatty liver disease,” “liver steatosis,” “exercise,” “physical activity,” “randomized controlled trial,” “gut microbiota,” and “gut metabolites.” Another search was performed on March 13, 2020, to find relevant NAFLD gene variations. The exact search strings can be found in Appendix A.

### 2.3. Inclusion and Exclusion Criteria

Articles were included if they met the following inclusion criteria: (1) RCTs in the adult population (age ≥ 18 years) that were written in English; (2) study subjects diagnosed with NAFLD through a biopsy, US, magnetic resonance imaging/magnetic resonance spectroscopy (MRS), or computed tomography; (3) studies with any sort of supervised physical activity for at least four weeks; (4) no diet modification in both the exercise intervention and control groups. The considered outcomes of interest were changes in IHL, stiffness, liver enzymes or other clinical parameters, anthropometrics, gut microbiota, and gut metabolites. Furthermore, there were no restrictions based on ethnicity, gender, and nationality.

The exclusion criteria were (1) uncontrolled, prospective, and cross-sectional studies; (2) unsupervised exercise interventions; (3) physical activity rates estimated using physical activity questionnaires; (4) review articles, case reports, conference abstracts, letters to editors, animal studies, and pilot studies.

### 2.4. Data Extraction and Quality Assessment

First, from the list of retrieved manuscripts, the titles and abstracts were independently screened by Ambrin Farizah Babu (A.F.B.), Susanne Csader (S.C.), and Johnson Lok (J.L.) based on the set of inclusion and exclusion criteria using RAYYAN [27] (Figure 1). All studies that did not fit into the chosen inclusion and exclusion criteria were excluded. Full-text articles that were retained from this stage were independently reassessed by the same authors. Any disagreements and discrepancies between the authors were resolved via discussion, debate, and consensus. If a conclusion was not reached, the final decision was made in consultation with Ursula Schwab. The primary outcomes were the liver parameters (IHL, AST, and ALT) and the secondary outcomes were the lipid parameters (HDL-C, LDL-C, TC, and TG), glucose parameters (fasting glucose, fasting insulin, and HbA1c), and anthropometrics (body mass index (BMI), weight, waist circumference, and body composition). The determination of quality was performed independently by S.C. and J.L. via the Cochrane risk-of-bias tool for randomized trials (RoB)2 tool (version August 2019) [28].

### 2.5. Data Analyses

#### 2.5.1. Data Preparation

To make all values across a particular parameter comparable, the data available in varying units were converted into one identical unit using appropriate conversion factors. HDL-C, LDL-C, and TC data were converted from milligrams per deciliter to millimoles per liter using the conversion factor 0.02586. For the conversion of TG data from milligrams per deciliter to millimoles per liter, a conversion factor of 0.01129 was used. A conversion factor of 0.0555 was used to convert milligrams per deciliter values of fasting glucose to millimoles per liter. Insulin data in picomoles per liter were converted to micro-international units per milliliter using a conversion factor of 6.945. HbA1c data from millimoles per mol were converted to a percentage (National Glycohemoglobin Standardization Program) values using an online HbA1c calculator (https://www.hba1cnet.com/hba1c-calculator/) (accessed on 11 December 2020). Thereafter, changes in the mean and standard deviation before and after the intervention were calculated for every parameter separately according to the Cochrane Handbook for Systematic Reviews of Interventions [29]. Briefly, for all paired analyses, the mean difference was calculated by subtracting the mean before and after the intervention. Missing standard deviations were imputed with a correlation coefficient of 0.8, representing the correlation between the baseline and follow-up scores. Thereafter, the change in standard deviation was calculated according to the Cochrane Handbook for Systematic Reviews of Interventions [29]. The original authors of the included studies were contacted if we did not have enough data for performing the statistical analyses.

#### 2.5.2. Statistical Analysis

To investigate the anthropometry changes after the exercise intervention, an unpaired two-tailed *t*-test was used.

#### 2.5.3. Meta-Analysis

All meta-analyses in this study were performed using R programming software (version 4.0.3.) and the packages “dmetar,” “meta,” “metafor,” and “metaviz.” Cumulative meta-analyses were used to explore the effects of exercise on NAFLD-related liver parameters (IHL, ALT, and AST), plasma lipid profile parameters (HDL-C, LDL-C, TC, and TG), and glucose metabolism parameters (fasting glucose, fasting insulin, and HbA1c). As all the outcomes were continuous, the standardized mean difference (SMD) was used as the measure for the effect size and was presented as a 95% confidence interval (CI) of the SMD. The SMDs of these parameters, from the baseline to the endpoint between groups (exercise vs. controls) in all studies, were calculated and pooled using random-effects or fixed-effect models. The heterogeneity across studies was tested using I^2^ statistics. The meta-analysis was based on a random-effects model if there was moderate to high variation (I^2^ > 25%); otherwise, a fixed-effects model was used.

The results for each of the parameters were represented as forest plots for overall exercise, along with subgroup analyses. The overall exercise included all the studies that measured the clinical parameter. The subgroup analyses were based on the exercise regimen (aerobic exercise/resistance training/combination of aerobic exercise and resistance training).

#### 2.5.4. Detection of Bias

Potential publication bias was assessed using funnel plots. The asymmetry of the funnel plots was assessed by using trim and fill imputation and Egger’s regression. Furthermore, a visual inference testing procedure using the line-up protocol with funnel plots was also employed [30]. However, restricting the search to only English-language articles added a language bias to our study.

#### 2.5.5. Outlier Detection

Potential statistical outliers were detected according to Viechtbauer and Cheung (2010) [31]. A study was considered as an outlier if the study’s CI did not overlap with the CI of the pooled effect [31]. Briefly, outliers with extremely small effects were detected by identifying all studies for which the upper bound of the 95% CI was lower than the lower bound of the pooled effect CI. Studies in this meta-analysis with extremely large effects were detected by searching for all studies where the lower bound of the 95% CI was higher than the upper bound of the pooled effect CI.

#### 2.5.6. Influence Analysis

Studies in this meta-analysis exerting a high influence on the overall results were identified by conducting an influence analysis based on the leave-one-out method [31]. Briefly, the results of the meta-analysis were recalculated k − 1 times, each time leaving one study out. The results were represented as Baujat plots, forest plots (sorted by effect size and heterogeneity, respectively), and influence plots, measuring the standard residual, DIFFITS value, Cook’s distance, covariance ratio, tau-squared, Q-value, hat value, and the weight of each individual study.

## 3. Results

### 3.1. Search Results

The electronic search (until 13 February 2020) yielded 1949 potentially relevant studies. After removing duplicates and screening through article titles and abstracts, 73 studies were retained for full-text reading. After excluding 61 ineligible studies based on the exclusion/inclusion criteria, 12 RCTs remained for review. Two more studies were excluded after there was no response from the authors for raw data to be included in the meta-analysis [32,33]. Finally, 10 studies were included for the meta-analysis (Table 1). A flow diagram including all stages is presented in Figure 1. One out of the 10 studies had two exercise intervention arms with different exercise regimes—aerobic exercise and resistance training [34]—which were counted as separate RCT exercise intervention arms. Each intervention arm was compared with the control arm and divided into its subgroups (aerobic exercise alone—six RCTs, resistance training alone—three RCTs, and a combination of aerobic exercise and resistance training—two RCTs). In total, 316 individuals with a mean age ranging from 39.7 to 62 years (from both exercise and control groups) were included in the analysis. This included 136 and 192 participants in the exercise and control groups, respectively (Table 2). None of the included studies had subjects with ischemic heart disease, whereas two of the included studies recruited participants with type 2 diabetes (T2D) that was treated with metformin [19,35]. However, no information was available about the number of participants with T2D. Details about other concomitant disorders were not available. Some studies had food records, but no diet results were reported [36,37,38,39,40]. All studies stated no diet changes. In addition, neither the search of gut microbiota and metabolites nor relevant NAFLD gene variations resulted in any studies for this review.

Anthropometry changes after the exercise intervention were analyzed using an unpaired two-tailed *t*-test. Of all the anthropometry changes reported in the studies, the body weight, whole-body fat mass, body fat percentage, lean body mass, BMI, and visceral adipose tissue were all non-significantly changed after the different exercise interventions (*p* > 0.05). The results from only one study [38] showed that the aerobic exercise intervention significantly reduced the waist circumference (*p* = 0.01), while the others remained non-significantly changed (*p* > 0.05).

There was variability in the number of participants, exercise duration, and settings provided in the interventions. The median (minimum–maximum) sample size was 12 (11–33) and 11.5 (6–31) for the intervention and control groups, respectively. The studies included in the analysis enrolled between 6 and 33 participants in each arm. Collectively, the exercise intervention length ranged from 8 to 16 weeks. The median duration was 12 weeks. The average exercise session ranged between 30 and 60 min. However, the frequency of exercise was as little as 2–3 times per week in eight studies to 4–5 times per week in the two other studies. The major forms of aerobic exercise studies were walking, cycling, and using a treadmill. For resistance training, full-body workouts were performed. The main study characteristics are shown in Table 2.

The results of the quality assessment for ALT are displayed in Figure 2. Overall, none of the studies were of high risk. Two studies showed some concerns for risk of bias in their randomization process as they did not provide details on the use of a random allocation sequence [19,34]. Due to a high dropout rate after randomization, four studies showed some concerns in their deviation from intended interventions, as well as missing outcome data [36,37,38,42]. In these studies, no sham exercise was adopted in the control groups and thus, participants, carers, and intervention administrators were not blinded and were aware of the participants’ assigned intervention. Despite these concerns for risk of bias, all studies were included in the subsequent meta-analysis.

### 3.2. Meta-Analysis

The meta-analysis was performed for liver parameters (IHL, ALT, AST), plasma lipid profile parameters (HDL-C, LDL-C, TC, TG), and glucose metabolism parameters (fasting glucose, fasting insulin, HbA1c). However, not all the included studies reported all of these clinical parameters; their numbers are shown in Table 1.

#### 3.2.1. Intrahepatic Lipid (IHL) Content

All the studies that measured IHL using MRS (*n* = 8) were included for the meta-analysis. Two other studies were not considered as they used US, which is not comparable with the results of MRS [34,40]. The results showed that IHL was significantly reduced upon exercise intervention compared with the controls (SMD: −0.76, 95% CI: −1.04, −0.48) (Figure 3a). Heterogeneity in the effect of overall exercise on IHL was not detected (*I^2^* = 0%, τ^2^ = 0, *p* = 0.53). Subgroup analyses of aerobic exercise only (SMD: −0.80, 95% CI: −1.14, −0.46) and a combination of aerobic exercise and resistance training (SMD: −0.80, 95% CI: −1.38, −0.22) showed a significant reduction in IHL compared with the controls. Heterogeneity was non-significant for aerobic exercise (*I^2^* = 21%, τ^2^ = 0.0416, *p* = 0.28) and the exercise combination (*I^2^* = 0%, τ^2^ = 0, *p* = 0.60). A subgroup analysis of resistance training was not possible as there was only one study included.

#### 3.2.2. Alanine Aminotransaminase (ALT)

The results of the meta-analysis showed that exercise can significantly reduce ALT (SMD: −0.52, 95% CI: −0.90, −0.14). Heterogeneity in the effect of overall exercise on ALT was high (*I^2^* = 60%, τ^2^ = 0.2299, *p* < 0.01). The results of the subgroup analyses of aerobic exercise showed a significant reduction in ALT (SMD: −0.63, 95% CI: −1.17, −0.10). However, resistance training alone (SMD: −0.33, 95% CI: −1.02, 0.37) and the exercise combination (SMD: −0.53, 95% CI: −1.92, 0.86) showed no improvements in ALT.

#### 3.2.3. Aspartate Aminotransaminase (AST)

The effect of exercise on AST was similar to that obtained for ALT (Figure 3c). AST was significantly reduced upon exercise intervention compared with the controls (SMD: −0.68, 95% CI: −1.21, −0.15). Heterogeneity in the effect of overall exercise on AST was high (*I^2^* = 77%, τ^2^ = 0.4925, *p* < 0.01). Subgroup analyses of aerobic exercise showed a significant reduction in AST (SMD: −0.73, 95% CI: −1.40, −0.07). However, the results of the resistance training alone (SMD: −0.12, 95% CI: −0.55, 0.31) and the combination of aerobic exercise and resistance training (SMD: −1.15, 95% CI: −3.69, 1.40) both showed no improvements in AST. Heterogeneity was significant for aerobic exercise (*I^2^* = 72%, τ^2^ = 0.4053, *p* < 0.01) and the exercise combinations (*I^2^* = 93%, τ^2^ = 3.1454, *p* < 0.01); it was non-significant for the resistance training alone (*I^2^* = 0%, τ^2^ = 0, *p* = 0.38).

#### 3.2.4. High-Density Lipoprotein Cholesterol (HDL-C)

HDL-C was slightly increased after the exercise intervention compared with the controls (SMD: 0.13, 95% CI: −0.17, 0.43) (Figure 4a). However, such a result was not statistically significant. Heterogeneity in the effect of overall exercise on HDL-C was very low and non-significant (*I^2^* = 13%, τ^2^ = 0.0194, *p* = 0.33). The results of the subgroup analyses of aerobic exercise alone (SMD: 0.09, 95% CI: −0.55, 0.73) showed no change in HDL-C. The results from the heterogeneity analysis of the aerobic exercise studies subgroup were also non-significant (*I^2^* = 47%, τ^2^ = 0.1523, *p* = 0.15). No analysis for the subgroup resistance training and the combination of aerobic exercise and resistance training was performed because there was only one study per group.

#### 3.2.5. Low-Density Lipoprotein Cholesterol (LDL-C)

As shown in Figure 4b, LDL-C was significantly decreased after exercise intervention compared with the controls (SMD: −0.34, 95% CI: −0.66, −0.02). Heterogeneity in the effect of overall exercise on LDL-C was non-significant (*I^2^* = 0%, τ^2^ = 0, *p* = 0.46). Subgroup analyses of aerobic exercise alone (SMD: −0.08, 95% CI: −0.57, 0.41) showed no change in LDL-C. Heterogeneity analysis on the aerobic exercise studies subgroup was also non-significant (*I^2^* = 0%, τ^2^ = 0, *p* = 0.59). No subgroup analysis was performed for resistance training only and the exercise combination because there was only one study per group.

#### 3.2.6. Total Cholesterol (TC)

TC was non-significantly decreased after exercise intervention compared with the controls (SMD: −0.46, 95% CI: −0.96, 0.04) (Figure 4c). Heterogeneity in the effect of overall exercise on TC was significantly high (*I^2^* = 68%, τ^2^ = 0.2961, *p* < 0.01). The results of the subgroup analysis of resistance training alone (SMD: −0.48, 95% CI: −0.92, −0.05) showed a significant decrease in TC, while both aerobic exercise alone (SMD: −0.00, 95% CI: −0.42, 0.41) and the exercise combination (SMD: −1.41, 95% CI: −3.10, 0.28) did not show a significant change in TC. Heterogeneity analysis on aerobic exercise alone (*I^2^* = 0%, τ^2^ = 0, *p* = 0.92) and resistance training alone (*I^2^* = 0%, τ^2^ = 0, *p* = 0.35) showed that both were non-significant, while that of the exercise combination was significant (*I^2^* = 85%, τ^2^ = 1.2692, *p* < 0.01).

#### 3.2.7. Triglyceride (TG)

TG was significantly decreased after exercise intervention compared with the controls (SMD: −0.59, 95% CI: −1.16, −0.02) (Figure 4d). Heterogeneity in the effect of overall exercise on TG was significantly high (*I^2^* = 79%, τ^2^ = 0.5815, *p* < 0.01). The results of the subgroup analysis of resistance training alone (SMD: −0.58, 95% CI: −1.02, −0.14) showed a significant decrease in TG, while aerobic exercise alone (SMD: −0.04, 95% CI: −0.37, 0.29) and the exercise combination (SMD: −2.50, 95% CI: −5.01, 0.00) both resulted in no change in TG. Heterogeneity analysis of aerobic exercise alone (I^2^ = 16%, τ^2^ = 0.0269, *p* = 0.32) and resistance training alone (I^2^ = 0%, τ^2^ = 0, *p* = 0.77) resulted in both being non-significant, while that of the exercise combination was significant (I^2^ = 90%, τ^2^ = 2.9249, *p* < 0.01).

#### 3.2.8. Fasting Glucose

Fasting glucose did not change after the exercise intervention compared with the controls (SMD: −0.21, 95% CI: −0.64, 0.22) (Figure 5a). A medium significant heterogeneity among all the studies was observed (I^2^ = 65%, τ^2^ = 0.2406, *p* < 0.01). The results of the subgroup analyses of aerobic exercise alone (SMD: −0.52, 95% CI: −1.09, 0.06), resistance training alone (SMD: −0.14, 95% CI: −1.10, 0.83), and a combination of aerobic exercise and resistance training (SMD: 0.33, 95% CI: −0.49, 1.15) all showed no change in fasting glucose.

#### 3.2.9. Fasting Insulin

As shown in Figure 5b, fasting insulin was not changed after an exercise intervention compared with the controls (SMD: −0.52, 95% CI: −1.13, 0.09). Heterogeneity in the effect of overall exercise on fasting insulin was significant (I^2^ = 81%, τ^2^ = 0.5951, *p* < 0.01). The results of the subgroup analysis of aerobic exercise alone (SMD: −0.22, 95% CI: −0.57, 0.13), resistance training only (SMD: 0.07, 95% CI: −0.36, 0.51), and the combination of aerobic exercise and resistance training (SMD: −2.42, 95% CI: −6.72, 1.88) all showed no change in fasting insulin. Heterogeneity analysis result of the aerobic exercise alone (I^2^ = 0%, τ^2^ = 0, *p* = 0.52) were zero and non-significant, while that of the combination was high and significant (I^2^ = 96%, τ^2^ = 0.4925, *p* < 0.01).

#### 3.2.10. Glycated Hemoglobin (HbA1c)

Figure 5c shows that HbA1c did not change after the exercise intervention (SMD: −0.23, 95% CI: −0.54, 0.07). Heterogeneity in the effect of overall exercise on HbA1c was non-significant (I^2^ = 0%, τ^2^ = 0, *p* = 0.88). The results of the subgroup analysis of aerobic exercise alone (SMD: −0.07, 95% CI: −0.57, 0.43) and resistance training alone (SMD: −0.39, 95% CI: −0.83, 0.05) both showed no change in HbA1c. Heterogeneity analysis of aerobic exercise alone (I^2^ = 0%, τ^2^ = 0, *p* = 0.71) and resistance training alone (I^2^ = 0%, τ^2^ = 0, *p* = 0.85) found that both were non-significant. No subgroup analysis was performed for the combination of aerobic exercise and resistance training because there was only one study in this group.

### 3.3. Detection of Bias: Funnel Pots

In meta-analyses, funnel plots are often used to assess potential publication bias. Visual inspection of the funnel plots can help to improve the objectivity and validity of funnel-plot-based conclusions by guarding the meta-analyst from interpreting patterns in the funnel plot that are plausible by chance [30]. In this meta-analysis, visual examination of the funnel plots showed no studies for HDL-C, LDL-C, and HbA1c; one study for IHL, TC, fasting glucose, and fasting insulin; two studies for ALT and TG; and three studies for AST that were indicated as having a potential publication bias (Figure 6a and Appendix A). However, visual inspections can lead to the drawing of incorrect conclusions [43].

Statistical tests, such as trim and fill and Egger’s regression, were suggested for assessing potential publication bias by establishing objectivity, while at the same time, controlling for type I errors [30,44,45]. In this study, based on Egger’s test, publication bias was not detected for the results on IHL (*p* = 0.92), AST (*p* = 0.07), HDL-C (*p* = 0.85), LDL-C (*p* = 0.75), TC (*p* = 0.64), TG (*p* = 0.25), fasting glucose (*p* = 0.33), and HbA1c (*p* = 0.88) (Figure 6a and Appendix A). For ALT (*p* = 0.19) (Appendix A) and fasting insulin (*p* = 0.03) (Appendix A), Egger’s test indicated the presence of a funnel plot asymmetry. However, these statistical tests focused on funnel plot asymmetry were quantified via the association of study effects with standard error [30]. Moreover, the power of the tests is lower when less than 10 studies are included [29].

The Cochrane Handbook for Systematic Reviews of Interventions [29] recommends not interpreting the statistical tests in isolation, but along with visual inspection of the funnel plot. Hence, to control for type I errors and to preserve the explorative nature of the visual inspection, a funnel plot of the observed data was simulated [30,46]. A line-up of 20 funnel plots was created, out of which, 19 showed data simulated under the null hypothesis and 1 with the observed data positioned randomly. In this study, the presence of heterogeneity was subject to visual inference. Hence, the fixed-effect model was used as the null. Moreover, to increase the power of the line-up procedure to detect small study effects, Egger’s regression line was drawn in each funnel plot, along with the trim-and-fill-imputed studies that were potentially missing due to publication bias.

After simulating the funnel plots, the position of the real data funnel plot was identified via visual inspection and confirmed by decrypting it using an R code. Based on the selected plot, it was observed that the included studies for HDL-C, LDL-C, and HbA1c did not indicate any potential publication bias, ALT and AST presented two studies, while IHL, TC, TG, fasting glucose, and fasting insulin presented one study each that indicated a potential bias (Figure 6 and Appendix A).

### 3.4. Outlier Detection

Statistical outliers were detected by identifying studies with extreme effect sizes [31]. No statistical outliers were detected for both fixed-effect and random-effects models for IHL, HDL-C, LDL-C, and HbA1c. One study (Shojaee-Moradie et al. 2016 [39]) was identified as the statistical outlier for both fixed-effect and random-effects models for AST (Appendix A), TC (Appendix A), TG (Appendix A), and fasting insulin (Appendix A). For the fixed-effect model for ALT, the study by Sullivan et al. (2012) [42] was identified as an outlier (Appendix A). Furthermore, the work by Cheng et al. (2017) [36] was identified as a statistical outlier for the fixed-effect models for AST (Appendix A) and fasting glucose (Appendix A).

### 3.5. Influence Analysis

A study detected as a statistical outlier may not be of much consequence if it exerts little influence on the results. However, some studies in a meta-analysis could exert a high influence on the overall results. As an example, even though the overall effect is non-significant, it could be that case that a highly significant effect is be found by removing one or a particular set of studies. Hence, influence analysis was performed on all parameters considered in this study to detect studies that mostly influence the overall estimates of the meta-analysis [31].

The influential analysis of IHL predicted the study by Cheng et al. [36] to be influential. This study, as seen from the Baujat plot, contributed considerably to the overall heterogeneity, as well as being very influential (Figure 7a). The extreme values in the diagnostic tests for influence measures identified this study as an influential case (Figure 7b). No conclusions of influential studies could be drawn based on the heterogeneity, as all I^2^ values remained zero when each of the studies was omitted (Figure 7d). However, the lowest overall effect favoring the exercise group was reached by omitting Cheng et al. [36], which again corroborated the finding that this study could be an influential case (Figure 7c). The study by Cuthbertson et al. [37] also moderately contributed to the overall heterogeneity and influence and omitting it had the highest overall effect, favoring the exercise group (Figure 7a,c). Nevertheless, this study did not report any extreme values in the diagnostic tests for influence measures (Figure 7b).

The influence analysis identified the study by Zelberg-Sagi et al. [40] as an influential case for ALT (Appendix A). This was shown in the Baujat plot (Appendix A). This plot also showed that the study conducted by Sullivan et al. [42] contributed highly to the overall heterogeneity, while at the same time, having a medium influence on the pooled results (Appendix A). In addition, the studies from Cuthbertson et al. [37] and Shojaee-Moradie et al. [39] showed moderate influences and moderate contributions to the overall heterogeneity (Appendix A). The highest effect favoring the exercise group could be reached after omitting the Zelberg-Sagi et al. study [40] (Appendix A). Omitting the Sullivan et al. study [42] resulted in having the least heterogeneity (Appendix A). However, this study did not have any extreme values in the diagnostic tests for influence measures (Appendix A).

Out of nine interventions that reported AST levels, four interventions were detected as influential cases—Cheng et al., Cuthbertson et al., Shojaee-Moradie et al., and Zelberg-Sagi et al. [36,37,39,40]. As shown in the Baujat plot, the studies by Cheng et al. [36] and Zelberg-Sagi et al. [40] highly contributed to influencing the pooled results (Appendix A). The study by Shojaee-Moradie et al. [39] moderately contributed to influencing the pooled results; however, it highly contributed to the heterogeneity (Appendix A). The study by Cuthbertson et al. [37] had a moderate influence on the overall results. These four studies had extreme values in the diagnostic tests for influence measures, and therefore, they were identified as influential cases (Appendix A). Furthermore, omitting the study by Zelberg-Sagi et al. [40] resulted in having the highest effect favoring the exercise group (Appendix A) and omitting the study by Shojaee-Moradie et al. [38] resulted in having the least heterogeneity (Appendix A).

The Baujat plot based on the influential analysis of HDL-C showed Sullivan et al. [43] contributed both to the overall influence and heterogeneity (Appendix A). However, no study had extreme values in the diagnostic tests for influential measures (Appendix A). Omitting this study resulted in the highest effect favoring the control and the least heterogeneity (Appendix A).

For LDL-C, the Baujat plot based on the influential analysis showed that the study by Cuthbertson et al. [37] had a high influence and a high contribution to overall heterogeneity (Appendix A). This study also had extreme values in the diagnostic tests for influential measures (Appendix A). Omitting this study resulted in the highest effect favoring the exercise group and the second least heterogeneity (Appendix A).

The Baujat plots for TC identified the study by Shojaee-Moradie et al. [39] as both influential and having contributed to the overall heterogeneity (Appendix A). In addition, Cuthbertson et al. [37] had extreme values in the diagnostic tests for influential measures (Appendix A). Omitting the Cuthbertson et al. study [37] resulted in having the highest effect favoring exercise (Appendix A) and omitting Shojaee-Moradie et al. [39] resulted in having the lowest effect favoring exercise (Appendix A). Furthermore, omitting Shojaee-Moradie et al. [39] made the overall results non-significant.

The Baujat plot results of TG showed that two studies—Cheng et al. and Shojaee-Moradie et al.—had a high influence (Appendix A) [36,39]. In addition to being highly influential, the Shojaee-Moradie et al. [39] study also contributed considerably to the overall heterogeneity (Appendix A). These two studies also had extreme values in the diagnostic tests for influence measures (Appendix A). Omitting these two studies resulted in having the least heterogeneity favoring the exercise group (Appendix A). Furthermore, upon calculating the pooled effects after omitting the study by Cheng et al., the highest effect was obtained (Appendix A).

For fasting glucose, three studies (Cheng et al., Shojaee-Moradie et al., and Zelberg-Sagi et al.) [36,39,40] were detected as influential, as shown in the Baujat plot (Appendix A). Cheng et al. [36] had the highest influence on the pooled effects and contributed to the highest overall heterogeneity. The study by Shojaee-Moradie et al. [39] had a moderate influence on the pooled effects and a moderate contribution to the overall heterogeneity. Based on its high sample size, Zelberg-Sagi et al. [40] were also identified as having an influence on effect size. All three studies had extreme values in the diagnostic tests for influential measures (Appendix A). Omitting the Cheng et al. [36] study resulted in the least effect favoring the control group. Omitting the studies by Zelberg-Sagi et al. [40] and Shojaee-Moradie et al. [39] resulted in the highest effect favoring the exercise group (Appendix A). Omitting these three studies resulted in the least heterogeneity (Appendix A).

Influence analysis of the fasting insulin parameters found that Shojaee-Moradie et al. [39] and Zelberg-Sagi et al. [40] were influential cases (Appendix A). Based on the Baujat plot, Shojaee-Moradie et al. [39] had a high influence and heterogeneity, and Zelberg-Sagi et al. [40] had a considerable influence on the pooled results (Appendix A). Omitting the Zelberg-Sagi et al. [40] study resulted in the highest effect favoring the exercise group and the second least heterogeneity (Appendix A). Omitting the study by Shojaee-Moradie et al. [39] resulted in the least effect favoring exercise (Appendix A). Furthermore, omitting this study made the overall results non-significant.

The Baujat plot based on the influential analysis for HbA1c showed that two studies—Cheng et al. and Zelberg-Sagi et al. [36,40]—were influential, with the highest influence on the pooled effects (Appendix A). However, these studies did not show any extreme values in the diagnostic tests for influential measures (Appendix A). Omitting Cheng et al. [36] resulted in the highest effect favoring the exercise group, while omitting Zelberg-Sagi et al. [40] resulted in the least effect favoring the exercise group (Appendix A). When each study was omitted, the I^2^ value remained zero; therefore, no conclusions regarding influential studies could be drawn based on the heterogeneity (Appendix A).

## 4. Discussion

This SR and meta-analysis analyzed the effects of different exercise regimens (without diet interventions) on the liver, plasma lipid profile, and glucose metabolism parameters in individuals with NAFLD. To the best of our knowledge, our study is the first SR and meta-analysis to investigate the effects of exercise alone on NAFLD-related clinical parameters, independent of weight loss.

Our findings showed that IHL was significantly reduced after the exercise interventions. Reduced levels of IHL are known to improve systematic inflammation and liver metabolic dysfunctions [47]. In addition, lower levels of NAFLD during the early stages of NAFLD could eventually prevent the disease progression to NASH.

The subgroup analyses also showed IHL reductions for both aerobic exercise alone and the combination of exercises. In addition to our study, many other SRs, which also included non-supervised studies, shorter resistance training sessions, longer aerobic interventions, and higher intensity aerobic interventions, reported similar IHL reductions [16,48,49,50,51]. For example, a reduction in IHL after aerobic and resistance training was reported by Baker et al. and Wang et al., respectively [49,50]. However, the conclusions of Baker et al. and Wang et al. were based on only one study that was included in the subgroup analysis. Moreover, unlike our study, which focused on the effects of exercise alone, Wang et al. reported a reduction in IHL in subjects undergoing a combined diet and exercise intervention. However, the actual mechanism(s) behind such an improvement remains uncertain. Multifactorial metabolic and molecular pathways for the pathogenesis of NAFLD were proposed [52,53]. One of the mechanisms involves insulin resistance (IR), which is one of the hallmarks of NAFLD. Systemic IR causes a disturbed suppression of adipose tissue lipolysis [54]. This leads to elevated free fatty acid (FFA) levels in the serum of NAFLD patients, which reach the liver [55,56]. In addition, glucose is converted to FFA via the de-novo lipogenesis in the liver that is caused by IR in skeletal muscles [54]. In the liver, these FFAs can either be synthesized to triglycerides and stored as lipid droplets in the liver leading to steatosis [57], excreted in very low-density lipoproteins (VLDL) [58], or oxidized in hepatic mitochondria (β-oxidation) [59]. It was shown that NAFL and NASH patients have increased fatty acid oxidation [60]. This β-oxidation increase may lead to mitochondrial dysfunction, and from this, reactive oxygen species (ROS) are formed and induce lipid oxidation, also called oxidative stress. Moreover, these created toxic metabolites can further contribute to mitochondrial damage [54]. Exercise intervenes in this process by improving the insulin resistance in adipose and muscle tissue [61], resulting in decreased systemic circulated FFA [62]. In addition, exercise downregulates the expression of several genes and proteins that are involved in lipogenesis, followed by reduced levels of FFA [54,63]. However, these findings were more documented in rodents than in humans [54].

Previous studies have shown that the duration of exercise intervention also affects the liver’s fat metabolism. For instance, a significant decrease in IHL (compared with the controls) was observed in two studies that conducted supervised aerobic exercise interventions for 8 and 12 weeks, respectively [64,65]. However, no change in IHL was reported by Shojaee-Moradie et al., who conducted an exercise intervention for six weeks [66]. However, this intervention was limited to only three sessions of 20 min per week. An eight-week resistance training intervention was also successful in reducing IHL [19]. Moreover, Bacchi et al. compared the effects of aerobic exercise or resistance training on IHL for four months and observed a 25–30% reduction in IHL from the baseline [67]. These results are consistent with the observations in this study, where it was found that both 12- and 16-week exercise interventions significantly reduced IHL (Appendix A). However, no conclusions could be drawn from the 8-week intervention as a meta-analysis could not be performed.

Elevated concentrations of liver enzymes, such as ALT and AST, often reflect non-specific liver damage or inflammation of liver cells [68]. This damage is more pronounced in NAFLD patients, making these enzymes appropriate markers for inflammation [68]. Other reasons for elevated liver enzymes include the use of certain medication and high-fat–high-carbohydrate diet consumption [69,70], which can be normalized without the influence of exercise [69,70]. In this study, the exercise intervention reduced ALT and AST slightly but significantly, which was consistent with Wang’s and Katsagoni’s meta-analyses [10,50]. In our study, aerobic exercise, but not resistance training nor the combination of exercises, reduced the concentrations of ALT and AST. Contradictory results were obtained by Keating’s meta-analysis, where shorter resistance training sessions were considered [16]. They found that IHL, but not ALT, was reduced after overall exercise. The limited inclusion of studies on aerobic exercise by Keatings et al. might have contributed to the contradictory results.

The main cause of death in NAFLD individuals is cardiovascular diseases (CVDs) [71]. One reason for developing CVDs is atherogenic dyslipidemia, which is characterized by a low HDL-C concentration and high LDL-C, TC, and TG concentrations [71]. Our analysis with five studies showed that HDL-C did not change after exercise, which is consistent with another SR. LDL-C was significantly decreased after exercise, but no changes were observed in aerobic exercise alone. Wang et al. (2020) reported a decrease after overall exercise and aerobic exercise [50]. Furthermore, one paper documented that high-intensity training (HIT) exercises have the greatest positive influence on LDL-C [72]. Moreover, they found that resistance training with increased repetitions/sets was better at reducing LDL-C levels than resistance training with high weights and a low number of repetitions [72]. These results suggest that LDL-C reduction might depend on the exercise regimen. However, only four papers were included for LDL-C, which cannot confirm any of the outcomes mentioned above. Thus, more exercise studies in NAFLD measuring LDL-C and HDL-C concentrations are warranted.

NAFLD patients have a disturbed cholesterol metabolism that is characterized by elevated cholesterol concentrations, increased cholesterol synthesis, diminished cholesterol absorption, and changed expression of cholesterol metabolism genes [73,74,75]. In this study, TC concentrations were not changed regarding overall exercise but were significantly changed in the resistance training subgroup. A similar non-significant change in TC concentration was confirmed by several other studies, irrespective of the exercise regime used [62,67,76]. In contrast, an exercise and diet intervention in NAFLD patients showed a >10% reduction of TC after exercise [77]. Interestingly, most of the measured cholesterol concentrations in these studies were already in the normal range. This could explain why the aerobic exercise subgroup showed no changes in TC concentrations.

Significantly reduced TG concentrations were noticed for the overall exercise effect and the subgroup for resistance training alone. The significant reductions in TG concentrations in the resistance training subgroup could be partly explained by exercise-induced myokine production, such as irisin, which was negatively correlated with TG, TC, and intrahepatic triglycerides [78,79]. Irisin increases total energy expenditure and modulates lipid metabolism by inhibiting enzymes, such as sterol regulatory element-binding protein-1c and fatty acid synthases in hepatocytes [80]. Irisin decreases in NAFLD patients [78] but can increase in obese people after resistance training, though not with aerobic training [81]. This mechanism could plausibly explain the increase in TG in the resistance training group. However, as irisin was not measured in any of the included studies, further validation was not possible.

In this study, exercise did not have an overall effect on glucose metabolism. The fasting glucose and insulin concentrations did not change after overall exercise and all subgroups, which is also in line with another meta-analysis [50]. Interestingly, Cheng et al., who recruited prediabetic NAFLD patients, reported significantly reduced fasting glucose levels in the exercise group compared with the control group [36]. Cuthbertson et al. and Shojaee-Moradie et al. reported reduced glucose concentrations in both the exercise and control groups [37,39]. In longer interventions, this reduction was no longer significant. This could be explained by a slight increase in physical activity (such as increased walking) in the control group receiving standard lifestyle advice.

Another glucose metabolism parameter, namely, HbA1c, which represents 2–3-month average glucose concentrations, did not change after overall exercise and subgroup exercise interventions. However, only five studies were included in this analysis. Nevertheless, our results were consistent with the previous findings. For example, Cheng et al. found a significant decrease in HbA1c in only the diet plus exercise intervention group but not the exercise-only group [36]. Houghton et al. found no significant decrease after exercise [41]. Additionally, a meta-analysis conducted by Chen et al. [82] also found no significant effect of resistance training on HbA1c. On the other hand, studies that were conducted in T2D subjects showed a significant HbA1c level reduction. One study by Church et al. reported that only a combination of resistance training and aerobic exercise leads to significantly decreased HbA1c levels [83], whereas another study by Yavari et al. found a significant reduction in the aerobic exercise group [84].

Current lifestyle interventions aiming for weight loss were the focus of clinical management of NAFLD treatment [13,85], where 5–10% weight loss is recommended for individuals with NAFLD and NASH [86]. In addition, several guidelines for NAFLD treatment, such as EASL, the American Association of Liver Disease, and the European Society of Clinical Nutrition and metabolism, recommend weight loss to improve NAFLD-related clinical parameters [9,87,88]. In this study, bodyweight loss was not significant after the exercise intervention. Additionally, several other anthropometry changes, such as whole-body fat mass, body fat percentage, lean body mass, BMI, and visceral adipose tissue, were non-significant after the exercise intervention for all the included studies. Only one study [38] showed a significant reduction in waist circumference. The beneficial effects of exercise, despite an absence of weight loss and anthropometry changes, could be attributed to several factors, including the increase in muscle strength, reduction in inflammation and oxidative stress, and changes in organokine concentrations [89].

Recent studies focusing on various risk factors of metabolic diseases have proposed that exercise, in addition to directly influencing metabolic responses, also contributes to a change in the gut microbiota and the composition of the metabolites they produce [90,91,92]. Furthermore, the relationship with NAFLD in humans has not yet been established. Therefore, one of the goals of our meta-analysis was to systematically search the literature for RCTs that investigate the effects of exercise on the NAFLD condition and change in gut microbiota. However, to the best of our knowledge, no study has reported such RCTs, and this research gap demands attention. Furthermore, our study indicated that the impact of exercise alleviating risk and symptoms related to NAFLD has so far not been addressed, with comprehensive metabolite profiling techniques holding the potential to bring out in-depth information related to endogenous metabolic differences and pathways that are responsible for modulating those factors. To understand the metabolic events and consequences behind NAFLD etiology and how exercise and other lifestyle factors may impact those, metabolomics techniques will be useful to accompany the traditional clinical parameters measured. Concerning this research gap, investigations of the relationship between exercise, NAFLD, and gut microbiota and the metabolites they produce are currently ongoing (ClinicalTrials.gov Identifier: NCT03995056). However, more research is needed in this field.

Our study has several strengths. The use of stringent criteria for selecting studies and data for the meta-analysis ensured more consistent and unbiased results. Only RCTs were included in our study, as they are the gold standard for assessing intervention effects, and randomization reduces selection bias [93], making them desirable for pooling results. Furthermore, all exercise regimens in the RCTs included in our study were performed under supervision. Studies where exercise compliance was only retrospectively reported by participants themselves were not considered. Another strength is the use of multiple methods to detect potential publication bias, statistical outliers, and influential cases, making our study less likely to be subjected to the effects of biases. Regarding data included in the meta-analysis, changes in IHL were only included when MRS was used as the quantification method. This study focused specifically on the effects of exercise (without diet intervention) on NAFLD-related clinical parameters. However, the effects of diet interventions or diet–exercise combinations should not be completely neglected.

Some limitations were also identified in our study. The limited number of studies included made it difficult to conduct meta-analyses on the subgroups. Out of the 10 included studies, not every study measured all the different NAFLD-related parameters considered in this study, and in some cases, some parameters were only available in two studies. Moreover, only two of the included studies conducted a combination of aerobic and resistance training. Therefore, a conclusion based on the meta-analysis in these subgroups with few studies might not reflect the actual effect. Furthermore, the intervention durations were only between 8 and 16 weeks, and thus, the long-term effects of exercise on NAFLD-related clinical parameters could not be determined. Only studies in the adult population were included; therefore, the results cannot be generalized to children or adolescents. Regarding statistical analyses, imputations were made for several studies for calculating the effect size to provide overall effect estimates, which involved making assumptions and may not reflect the actual data. Moreover, only studies in the English language were included. Furthermore, the heterogeneity for most of the outcomes, including ALT, AST, TC, TG, fasting glucose, and fasting insulin, was high [94]. This could have possibly arisen due to clinical differences, methodological issues from the original study, including randomization, use of absolute rather than relative measures of risk, and publication bias [95]. Furthermore, the different subgroup analyses that had a different true effect might also have contributed toward the heterogeneity.

## 5. Conclusions

Based on our meta-analysis, we concluded that exercise overall likely had a beneficial effect on alleviating NAFLD without significant weight loss. After overall exercise, IHL, ALT, AST, LDL-C, and TG were reduced. The aerobic exercise seemed to alleviate NAFLD-related liver parameters (IHL, ALT, and AST), while resistance training was more effective at alleviating TG and TC. The combination of aerobic and resistance training alleviated only IHL. These results could allow for the establishment of exercise (or lifestyle modification) guidelines, where weight loss is not the focus but could be a consequence of the lifestyle modification in some patients. However, it must be noted that our conclusions are based on only a small number of studies with small sample sizes. Therefore, more studies are warranted for a comprehensive and meaningful comparison of different exercise regimes, the long-term effects of exercise, and the gut microbiota and metabolites’ contributions.

## Figures and Tables

**Figure 1 nutrients-13-03135-f001:**
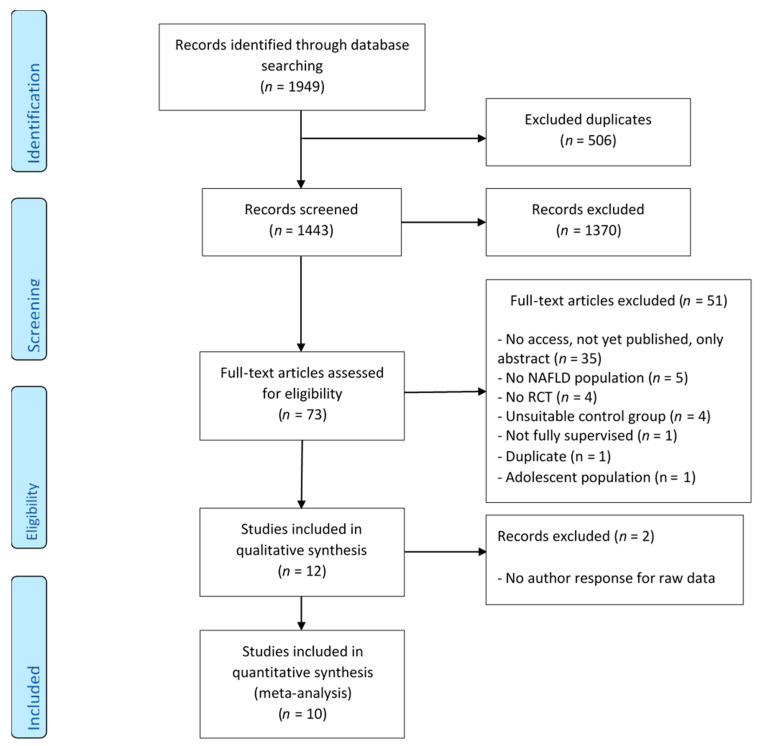
Modified PRISMA flow chart diagram showing the process of study selection [26]. NAFLD: non-alcoholic fatty liver disease; RCT: randomized control trial.

**Figure 2 nutrients-13-03135-f002:**
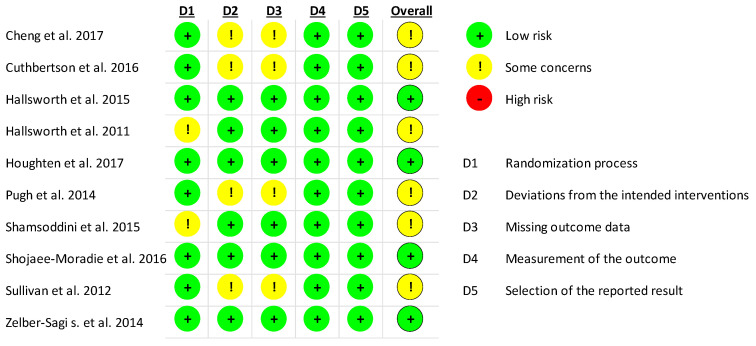
Quality assessment for ALT of all included studies.

**Figure 3 nutrients-13-03135-f003:**
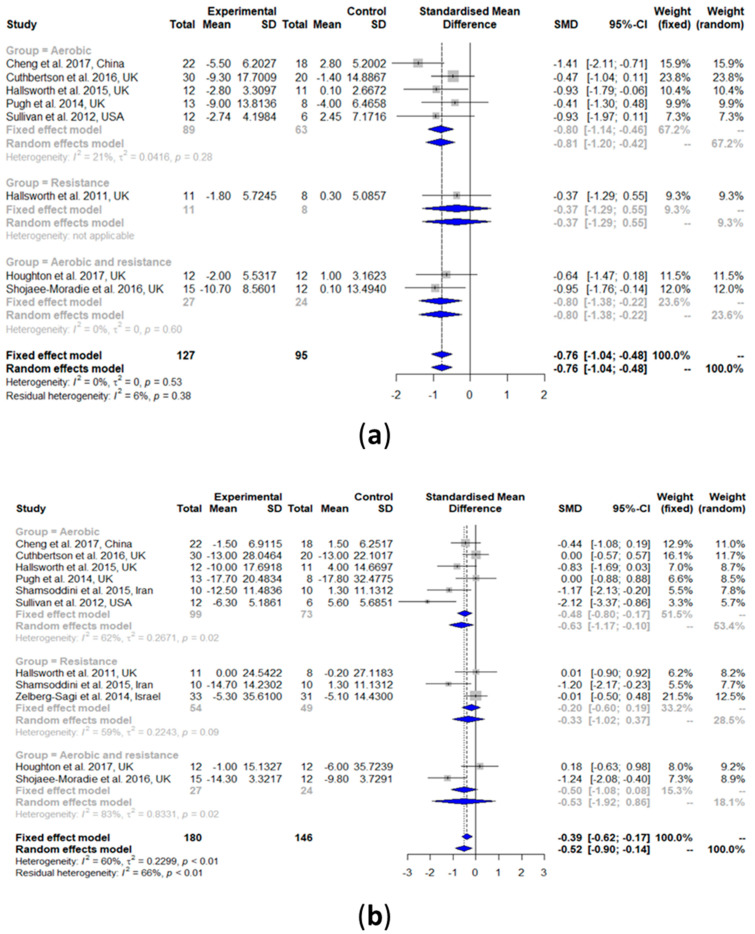
Forest plots of the effect of exercise on liver parameters: (**a**) intrahepatic liver content (IHL), (**b**) alanine transaminase (ALT), and (**c**) aspartate transaminase (AST). SD: standard deviation; SMD: standardized mean difference; CI: confidence interval.

**Figure 4 nutrients-13-03135-f004:**
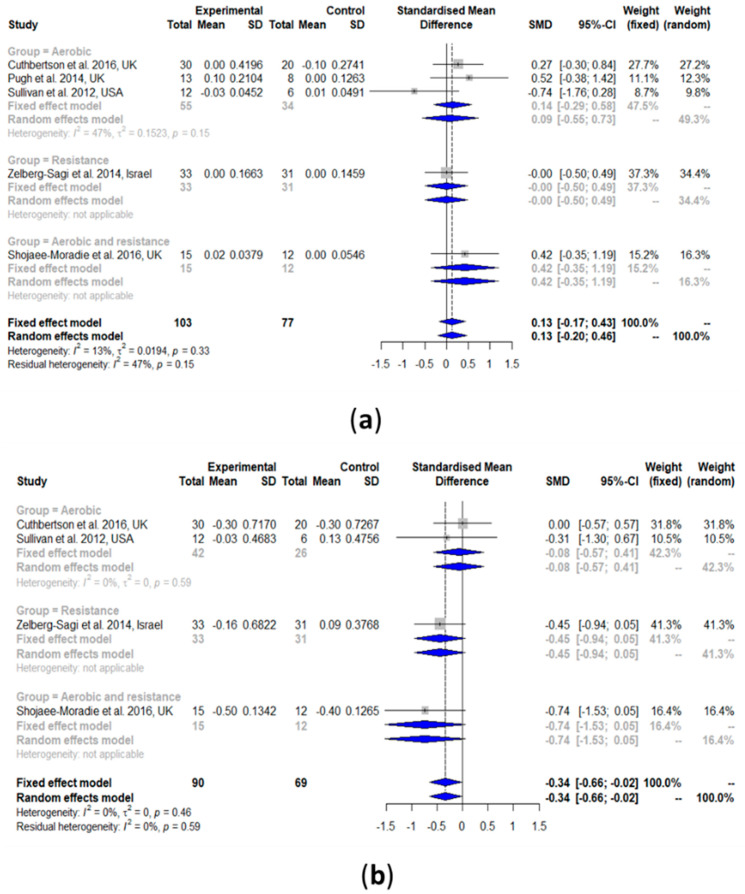
Forest plots of the effect of exercise on lipid metabolism parameters: (**a**) high-density lipoprotein cholesterol (HDL-C), (**b**) low-density lipoprotein cholesterol (LDL-C), (**c**) total cholesterol (TC), and (**d**) triglyceride (TG). SD: standard deviation; SMD: standardized mean difference; CI: confidence interval.

**Figure 5 nutrients-13-03135-f005:**
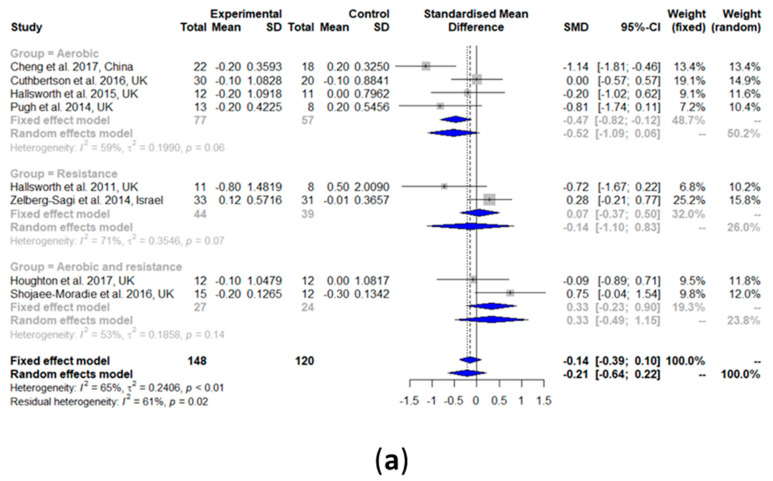
Forest plots of the effect of exercise on glucose metabolism parameters: (**a**) fasting glucose, (**b**) fasting insulin, and (**c**) glycated hemoglobin (HbA1c). SD: standard deviation; SMD: standardized mean difference; CI: confidence interval.

**Figure 6 nutrients-13-03135-f006:**
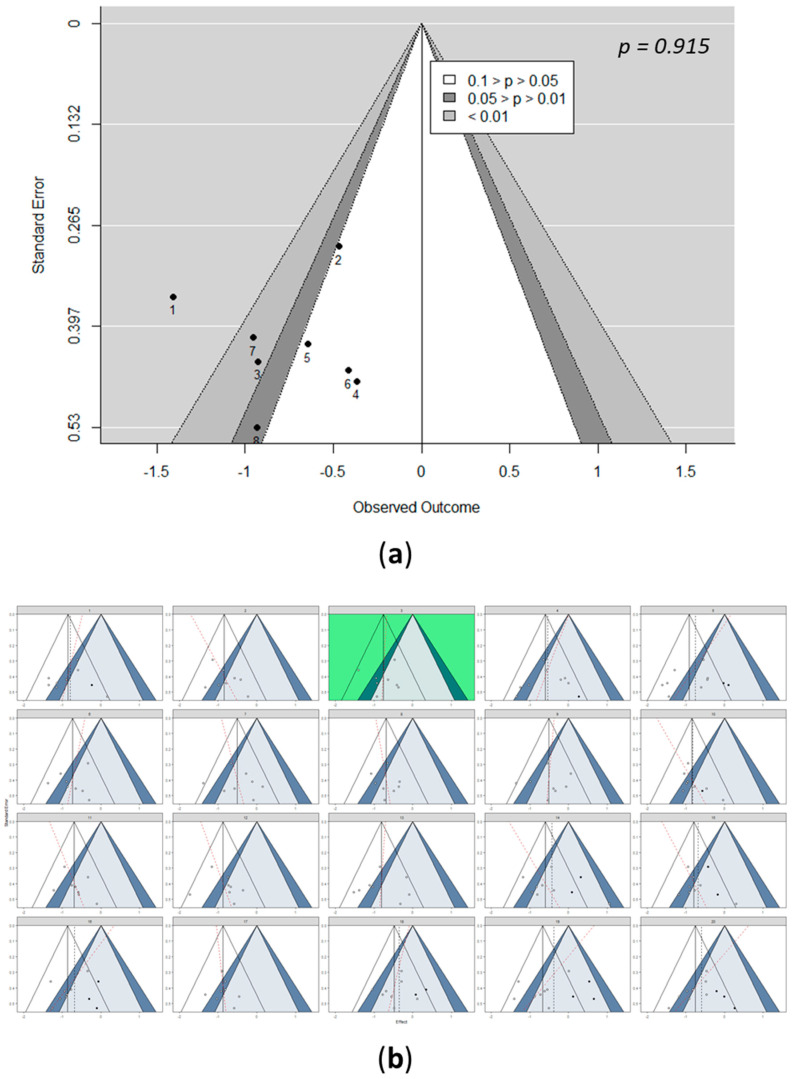
Detection of bias for IHL. (**a**) Funnel plot of studies included in the IHL meta-analysis, including 95% confidence contours and significance contours at 0.05 and 0.01 levels, with Egger’s test for the detection of publication bias of included studies in the IHL meta-analysis. (**b**) Funnel plot line up. One funnel plot shows the actually observed data (green), while the other 19 funnel plots were simulated under the null hypothesis of a fixed-effect meta-analytic model. Shown are 95% confidence contours, the summary effect (vertical line), and significance contours at the 0.05 and 0.01 levels. The black dotted lines and the red lines represent the trim and fill and Egger’s regression, respectively. See Appendix A in the Appendix A for the detection of bias for all included parameters.

**Figure 7 nutrients-13-03135-f007:**
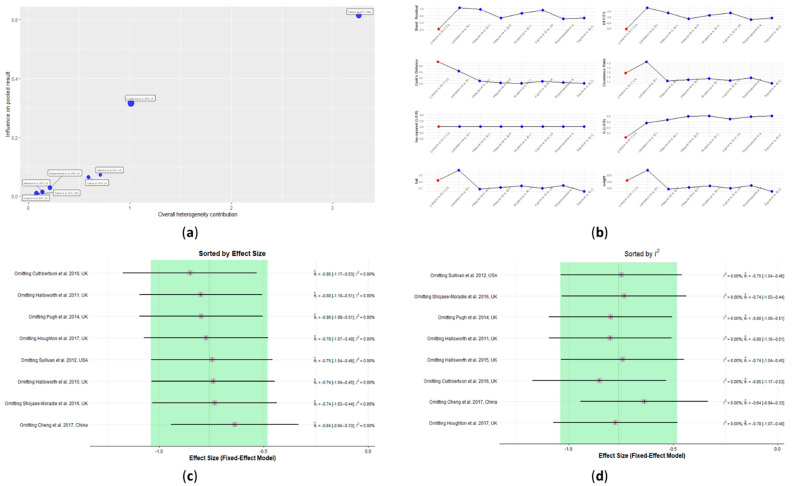
Influence analysis plots for intrahepatic liver content (IHL): (**a**) Baujat plot, (**b**) diagnostic tests for influence measures, (**c**) leave-one-out analyses sorted by decreasing order of effect size, and (**d**) leave-one-out analyses sorted by decreasing order of heterogeneity. See Appendix A in the Appendix A for influence analysis plots for all included parameters.

**Table 1 nutrients-13-03135-t001:** Included studies for the meta-analysis for each clinical parameter.

Clinical Parameter	Aerobic Exercise	Resistance Training	Exercise Combination (Aerobic Exercise + Resistance Training)	Overall Exercise (Aerobic Exercise + Resistance Training + Exercise Combination)
IHL	5	1 *	2	8
Liver stiffness	-	-	1 *	1 *
ALT	6	2	2	10
AST	5	2	2	9
HDL-C	3	1 *	1 *	5
LDL-C	2	1 *	1 *	4
TC	3	2	2	7
TG	5	2	2	9
Fasting glucose	4	2	2	8
Fasting insulin	4	2	2	8
HbA1c	2	2	1 *	5

IHL: intrahepatic lipid content; ALT: alanine aminotransaminase; AST: aspartate aminotransaminase; HDL-C: high-density lipoprotein cholesterol; LDL-C: low-density lipoprotein cholesterol; TC: total cholesterol; TG: triglyceride; HbA1c: glycated hemoglobin. Overall exercise includes all the studies that measured clinical parameters. * No meta-analysis could be performed.

**Table 2 nutrients-13-03135-t002:** Main study characteristics.

First Author, Year of Publication, and Country	Sample Size (M/F)	Mean Age (SD) (Years)	Disease Stage	Techniques Used to Assess the Liver Fat	Exercise Intervention (Aerobic/Resistance/Combination)	Training Sessions/Week	Intervention Duration (Weeks)	Food Diary (Y/N)
Cheng et al. 2017, China [36]	C: 18 (4/14) I: 22 (5/17)	C: 60 (3.4) I: 59 (4.4)	NAFLD with impaired FG	1H-MRS	Aerobic: Nordic brisk walking + stretching and other group exercise increased VO2max from 60% to 75%	2–3 times 30–60 min	12	Y
Cuthbertson et al. 2016, UK [37]	C: 20 (16/4) I: 30 (23/7)	C: 52 (46,59) * I: 50 (46,58) *	NAFLD	1H-MRS	Aerobic: moderate (30% HRR), treadmill, cross-trainer, bike ergometer, rower, progressing weekly based on HR responses (60% HRR by week 12)	3 times for 30 min up to 5 times for 45 min	16	Y
Hallsworth et al. 2015, UK [35]	C: 11 I: 12	C: 52 (12) I: 54 (10)	NAFLD	1H-MRS	Aerobic: cycle-ergometer-based HIIT	3 times for 30–40 min	12	N
Hallsworth et al. 2011, UK [19]	C: 8 I: 11	C: 62 (7.4) I: 52 (13.3)	NAFLD	1H-MRS	Resistance: bicep curl, calf raise, triceps press, chest press, seated hamstring curl, shoulder press, leg extension, and lateral pull-down	3 times for 45–60 min	8	N
Houghton et al. 2017, UK [41]	C: 12 I: 12	C: 51 (16) I: 54 (12)	NASH	1H-MRS biopsy	Aerobic and resistance: cycling + resistance training with weight (hip and knee extension, horizontal row, chest press, vertical row, knee extension)	3 times per for 45–60 min	12	N
Pugh et al. 2014, UK [38]	C: 8 (4/4) I: 13 (7/6)	For all: 48 (44.51)	NAFLD, obese	1H-MRS	Aerobic: combination of treadmill and cycle ergometer-based exercise 30% HRR to 45% HHR to 60% HRR	3 times per for 30 min up to 45 min	16	Y (C: N)
Shamsoddini et al. 2015, Iran [34]	C: 10 (10/0) I1: 10 (10/0) I2: 10 (10/0)	C: 45.8 (7.3) I1: 39.7 (6.3) I2: 45.9 (7.3)	NAFLD	US	Group 1—Aerobic: running on treadmill, 60% MHR to 75% MHR Group 2—Resistance: triceps press, bicep curl, calf raise, leg press, leg extension, lat pull down, sit-ups	3 times for 45 min	8	N
Shojaee-Moradie et al. 2016, UK [39]	C: 12 (12/0) I: 15 (15/0)	C: 52.8 (3) I: 52.4 (2.2)	NAFLD	1H-MRS	Aerobic and resistance: gym based/outdoor aerobic + resistance training moderate intensity (40–60% HRR)	4 to 5 times for 20 min up to 60 min	16	Y
Sullivan et al. 2012, USA [42]	C: 6 (1/5) I: 12 (4/8)	I: 48.6 (2.2) C: 47.5 (3.1)	NAFLD obese,	MRS	Aerobic: brisk walk on treadmill, 45–55% VO2peak	5 times for 30–60 min	16	N
Zelberg-Sagi et al. 2014, Israel [40]	C: 31 (18/13) I: 33 (16/17)	C: 46.6 (11.4) I: 46.3 (10.32)	NAFLD	US	Resistance: leg press/extension/curl, seated chest press/rowing, latissimus pull down, bicep curl, shoulder press	3 times per for 40 min	12	Y

C: control; I: intervention; SD: standard deviation; NASH: non-alcoholic steatohepatitis; NAFLD: non-alcoholic fatty liver disease; FG: fasting glucose; GT: glucose tolerance; HR: heart rate; HRR: heart rate reserve; VO2max: max. oxygen uptake; 1H-MRS: proton magnetic resonance spectroscopy; US: ultrasound; M: male; F: female; N: no; Y: yes; * range.

## Data Availability

Access to the data, R codes, and/or material can be sought via contacting the responsible authors.

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
