# Peer review of "Positive Effects of Exercise Intervention without Weight Loss and Dietary Changes in NAFLD-Related Clinical Parameters: A Systematic Review and Meta-Analysis"

_nutrients, 2021, doi:10.3390/nu13093135_

Round 1
Reviewer 1 Report
Nil further
Author Response
Thank you for your comment,
We had our manuscript professionally edited by the MDPI's English editing service.
Reviewer 2 Report
Thank you very much for the opportunity to evaluate this article.
The manuscript deals with an important topic to evaluate the positive effects in NAFLD subjects upon exercise intervention without significant weight loss and changes to their diet.
The manuscript is very well written in English, and this review topic is of great clinical importance.
I suggest the authors consider the following comments to improve the quality of the manuscript.
Major comments
- This review did not fully follow the PRISMA statements. Please review and complete the PRISMA checklist and show the PRISMA checklist.
- The conclusions did not take into account the certainty of the evidence on the outcomes.
Minor comments
Abstract
- Please specify the information sources (e.g. databases, registers) used to identify studies and the date when each was last searched.
- Please specify the methods used to assess risk of bias in the included studies.
- The authors should provide a brief summary of the limitations of the evidence included in the review (e.g. study risk of bias, inconsistency and imprecision).
- Please specify the primary source of funding for the review.
- Please provide the register name and registration number.
- The authors should state the conclusion with the certainty of the evidence.
Materials and methods
- The authors should clarify the primary and secondary outcomes. In the protocol, lipids parameters were secondary outcomes, and there was no mention of weight, body composition, weight circumference, or BMI in the Methods section.
- Reference 26 has been updated to the PRISMA 2020 Statement (BMJ. 2021;372:n71.).
- Although the authors consider publication bias in a variety of ways, but restricting the search to English-language articles may be one of the limitations.
- In the meta-analysis, it was methodologically incorrect to use a fixed effects model when there was moderate to high variability (I2 > 25%). The decision to use a fixed-effects model is based on clinical, not statistical, differences. The studies included in this review had clinical differences and therefore should use a random-effects model.
- The authors should describe any methods used to assess certainty of the evidence for an outcome.
- Please mention that the authors have asked the original authors for missing data.
Results
- Please provide references for the two studies in the following sentences: “Two more studies were excluded after no response from the authors for raw data to be included in the meta-analysis.” (on page 4, lines 186-187, in the methods section)
- Risk of bias assessments should be performed for each outcome according to the Cochrane handbook. Please show RoB2 assessments for each outcome.
- Please reevaluate assessment of domain 2 for Risk of Bias.
- The authors should evaluate the certainty of the evidence according to GRADE approach for each outcome.
Discussion
- The authors should discuss what the clinical implications of a decrease in IHL are.
- The authors need to discuss the reasons for the studies that caused the high heterogeneity for each outcome.
Conclusions
- The conclusions should be stated with the certainty of the evidence regarding the outcome. “Our meta-analysis concludes that exercise overall may or likely have a beneficial effect on alleviating NAFLD without significant weight loss.”
Author Response
Dear reviewer,
Thank you for your comments. Please find the attached document.

Round 2
Reviewer 2 Report
The responses to the comments were well-addressed and the manuscript was improved by revision. However, one minor revision is needed.
Minor revision
The deletion of Figure 2 was not kind to the readers. Please show figure 2 as a RoB 2 for ALT. The authors need to select which specific result from the included trials to assess based on the Cochrane handbook (https://training.cochrane.org/handbook/current/chapter-08#section-8-1).
“The results of the quality assessment for ALT are displayed in Figure 2.” in lines 257 and re-show Figure 2.
“8.2.1 Selecting which results to assess within the review#section-8-2-1
Before starting an assessment of risk of bias, authors will need to select which specific results from the included trials to assess. Because trials usually contribute multiple results to a systematic review, several risk-of-bias assessments may be needed for each trial, although it is unlikely to be feasible to assess every result for every trial in the review. It is important not to select results to assess based on the likely judgements arising from the assessment. An approach that focuses on the main outcomes of the review (the results contributing to the review’s ‘Summary of findings’ table) may be the most appropriate approach (see also Chapter 7, Section 7.3.2).”
Author Response
Dear Reviewer,
Thank you very much for this comment. It was a mistake from our side for accidentally deleting Figure 2. Our sincere apologies.
We have now re-shown figure 2. And rewritten the title of the figure “Figure 2. Quality assessment for ALT of all included studies.” (line 253) and the description of the results “The results of the quality assessment for ALT are displayed in Figure 2.” (line 242).
This manuscript is a resubmission of an earlier submission. The following is a list of the peer review reports and author responses from that submission.
Round 1
Reviewer 1 Report
Ambrin Farizah Babu et al., in the manuscript entitled “Positive effects of exercise intervention without weight loss and dietary changes in NAFLD-related clinical parameters: A systematic review and meta-analysis”, aimed to distinguish the contribution of different exercise regimens to the different clinical parameters. The main topic of the manuscript is very interesting, the authors made an effort (especially from the statistical point of you) to present and support their findings. However, the proposed results need to be improved since there are several inconsistencies. Overall, the manuscript needs to be improved and it should be made more readable. In the present form it cannot be accepted for the publication. I do hope that the authors will find helpful my comments below in order to improve their interesting manuscript.
Major comments:
- Abstract: authors stated that: “No studies investigating the role of gut microbiota and exercise in NAFLD were found.” Is there need to mention it?
- Introduction should be summarized and more focused on the specific aspects of the role of exercise (lifestyle changes) in the treatment of NAFLD.
In addition, the authors stated: “Exercise alone (with or without weight loss) has been shown to improve clinical parameters of NAFLD such as…”, but at the end they said: “The novelty of this review was the findings of positive effects in NAFLD subjects upon exercise intervention without weight loss.” This is somewhat confusing. This is the systematic review and meta-analysis, so I do not think that this statement is correct. Please consider to reword it.
- One of inclusion criteria was “study subjects diagnosed with NAFLD through biopsy, ultrasonography (US), magnetic resonance imaging (MRI) /magnetic resonance spectroscopy (MRS) or computed tomography”. Do authors think that this criterion is too general and that several gap can be found if different approaches have been used to diagnose NAFLD? It should be discussed.
- The authors stated that, in total, 316 individuals (from both exercise and control groups) were included. It should be important to present how many subjects were in each group. Does this mean that all 10 studies had a control group? What about the mean age of the participants? Could participants’ age influence the results and/or their compliance to the exercise interventions? What about concomitant disorders (such as diabetes mellitus type 2) that could influence significantly lipid levels?
- The authors stated: “All statistical analyses involving the meta-analysis in this study were performed using R programming software.” What do the authors mean by “all statistical analysis involving the meta-analysis”? All analyses should be related to this meta-analysis or not?
- The authors repeated several times the following sentence: “The meta-analysis was performed for the liver parameters (IHL, ALT, AST), plasma lipid profile parameters (HDL-C, LDL-C, TC, TG) and glucose metabolism parameters (fasting glucose, fasting insulin, HbAC1).” Please try to avoid repetition.
- It seems that all studies included in the present meta-analysis measured IHL and the liver enzyme ALT (as stated by the authors). Then, the authors stated the following: “…the liver enzyme AST was not analyzed in two studies [22,43]. HDL-C and LDL-C analysis were performed by five studies [41-45]. Furthermore, TG was not mentioned in one study [39] and TC in two studies [39,40]. Fasting glucose and insulin concentrations were not measured by two studies [39,43]. HbA1c analysis was conducted by five studies [22,38,40,45,46].”
So, can any conclusion be drawn? Please consider that some parameters were available only in 2 studies. This is a very important limitation that should be mentioned among the limitations.
Yet, the authors stated later: “Data on IHL was available from all studies. However, only those studies that reported IHL as percentage values (n=8) were considered for meta-analysis.” Please be careful and try to be precise as much as possible. There are several inconsistencies and it seems that the first statement (that 10 studies with 316 individuals were included), actually is not true. Please correct this accordingly.
- “Effect of all the exercise interventions on body weight change and fat mass was non-significant for all the studies. Therefore, significant changes in the NAFLD-related clinical parameters in exercise groups compared with control reported in our study were all independent of weight loss and fat mass reduction.” Did the authors consider fat mass reduction too? It has not been mentioned. I did not find any results reported on body weight or fat mass. Do the authors think that such changes can be correlated with changes in other parameters (e.g. lipids)?
- The authors stated: “For IHL, only studies using MRS as detecting hepatic fat were included in the meta-analysis.” Please clarify this point and explain why did you decide this, when previously you stated that IHL was available in all studies?
- I would suggest to present more clearly Forest plots in the results section, they are not readable. Overall, results section can be presented more clearly and shortly.
- One of the most common statement in the results section is: “Five, one, and two studies focused on aerobic exercise, resistance training, and a combination of these, respectively.” This point should be clearly stated in the material and methods section. Considering different types of exercises (aerobic exercise, resistance training, and their combination) how many studies are there?
- The authors stated: “Each intervention arm was compared with the control arm and divided into its subgroups (aerobic exercise alone - 6 RCTs, resistance training alone – 3RCTs, and a combination of aerobic exercise and resistance training– 2 RCTs).” Does this mean that “exercise overall” means all these subgroups together? I suggest to define the meaning of “exercise” or “exercise overall” in order to better understand what the authors discussed later about the changes after exercise (e.g. “Heterogeneity in the effect of exercise on ALT was high” (which one? overall exercise?)).
This point is very important as the aim was to distinguish the contribution of different exercise regimens to the different clinical parameters.
Additionally, did each arm has its own control? What were healthy status and lifestyle of the controls?
- Similarly, the authors stated: “The studies included in the analysis enrolled between 6 and 33 participants in each arm. The duration of the studies was between 8-16 weeks.” This should be present in details. The duration of exercise intervention is not clear (between 8 to 16 weeks, please report the mean). In addition, given that all parameters were not present in all studies included (10), it is so difficult to decide the mean duration, eventually it should be mentioned. Then, how could you compare 6 vs 33 participants? Pleas clarify.
- In discussion section please discuss your findings in more details. For instance, the authors stated: “These results were also confirmed by several other systematic reviews with different included studies”. Please discuss briefly what were the different inclusion criteria? What was different and what was similar to your study?
In general, compare more your findings with others’ findings instead of just mention it (e.g. contradictory results were obtained by Keating’s meta-analysis [53]. They found out that IHL, but not ALT, was reduced after overall exercises. The reason for such contradictory and inconsistent results may be attributed to the differences in the included studies, exercise regime, intensity, duration, and frequency). Please summarize what were the main differences in selection of the studies, exercise regime etc., and support your own findings. Overall, discussion should be shorter and more readable.
- The authors stated: “The significant reductions in TG concentrations in the resistance training subgroup could be partly explained by exercised-induced myokine production such as irisin, which is negatively correlated with TG, TC, and intra hepatic triglycerides [90,91].” The authors did not study exercised-induced myokine production such as irisin, so I am not sure this is appropriate to mention in the present form.
- This is one more example of the repetition through the manuscript: “Only IHL content quantified by MRS was used for performing the statistical analyses. Therefore, only 8/10 studies were included in the meta-analysis.” There are several repetitions, please check it carefully.
- Given that this meta-analysis did not include data on gut microbiota, there is no need to discuss widely about that. I suggest to make just a brief discussion what is missing in the literature and what is expecting to be reported in the field in the following years. Consequently, limits of the present study should be presented in more details.
- Do not the authors believe that the conclusions are general and limited on only few studies included? Considering that all parameters were not present in all studies, a final conclusion cannot be drawn (as also has been mentioned by the authors). Please try to reword this section.
- Please check carefully English language as well as the abbreviations through the text.
Author Response
Dear reviewer, thank you for your helpful comments. We have tried our best to revise the manuscript according to your suggestions and comments:
- Abstract: authors stated that: “No studies investigating the role of gut microbiota and exercise in NAFLD were found.” Is there need to mention it?
Our search terms included gut microbiota as one of the primary outcomes. Therefore, we think it is essential to mention it in the abstract.
- Introduction should be summarized and more focused on the specific aspects of the role of exercise (lifestyle changes) in the treatment of NAFLD.
We have shortened the introduction to have a better focus on the role of exercise in NAFLD treatment.
In addition, the authors stated: “Exercise alone (with or without weight loss) has been shown to improve clinical parameters of NAFLD such as…”, but at the end they said: “The novelty of this review was the findings of positive effects in NAFLD subjects upon exercise intervention without weight loss.” This is somewhat confusing. This is the systematic review and meta-analysis, so I do not think that this statement is correct. Please consider to reword it.
Thank you for pointing it out. Indeed, several previous studies have shown an improvement of NAFLD parameters after exercise without weight loss. Our study is the first systematic review and meta-analysis to look at it from this perspective. We have rephrased the text as follows:
“To the best of our knowledge, this systematic review (SR) and meta-analysis are the first to demonstrate the positive effects in NAFLD subjects upon exercise intervention without significant weight loss and changes to their diet.” (Page 2, lines 81 to 84 in the version without track changes; Page 3, lines 100 to 104 in the version with track changes)
- One of inclusion criteria was “study subjects diagnosed with NAFLD through biopsy, ultrasonography (US), magnetic resonance imaging (MRI) /magnetic resonance spectroscopy (MRS) or computed tomography”. Do authors think that this criterion is too general and that several gap can be found if different approaches have been used to diagnose NAFLD? It should be discussed.
Blood tests, imaging, and biopsy are commonly used to diagnose NAFLD. Most serological markers have not been validated in independent cohorts and therefore were not used as an inclusion criterion in our study. The imaging methods have been known to have high accuracies, and biopsy remains to be a gold standard. Although we used several approaches, they were used to merely ascertain if the participants had NAFLD or not. They were not used for any other comparison purposes except for IHL values. Here again, the IHL values from MRS (reported as liver fat percentage, %) were only used for the meta-analysis. There were two more studies which used ultrasound to report IHL values as hepatic fat grade or hepatorenal-ultrasound index (HRI) score. Therefore, the values from MRS and ultrasound cannot be compared and therefore they were excluded from the meta-analysis.
- The authors stated that, in total, 316 individuals (from both exercise and control groups) were included. It should be important to present how many subjects were in each group. Does this mean that all 10 studies had a control group? What about the mean age of the participants? Could participants’ age influence the results and/or their compliance to the exercise interventions? What about concomitant disorders (such as diabetes mellitus type 2) that could influence significantly lipid levels?
Thank you for your comment. We had previously indicated the number of subjects in each group along with their mean age in Table 1. We have now added these details in the result section (3.1) as follows: “In total, 316 individuals with a mean age ranging from 39.7 to 62 years (from both exercise and control groups) were included in the analysis. This included 136 and 192 participants in the exercise and control groups, respectively (Table 1).” (Page 4, lines 194 to 197 in the version without track changes; Page 5, lines 227 to 230 in the version with track changes)
In addition, as requested, we have also added the missing details about the concomitant disorders as follows: “None of the included studies had subjects with ischemic heart disease, whereas two of the included studies recruited participants with type 2 diabetes (T2D) treated with metformin [19,33]. However, no information was available about the number of participants with T2D. Details about other concomitant disorders were not available.” (Page 4, lines 197 to 200 in the version without track changes; Page 5, lines 230 to 235 in the version with track changes)
We only evaluated Randomised Control Trials (RCTs) for our study and it is a golden standard of studies related to health, diseases, and metabolism to include control groups. Hence, all the selected 10 studies had a control group included.
We do not think that the participants’ age could influence the results and/or their compliance to the exercise interventions. This is because we only included those studies that were supervised. The supervised interventions typically avoid any incorrect results that could be caused by non-compliance. Further, the compliance for all groups should be similar (identical) for all studies, independent of age.
- The authors stated: “All statistical analyses involving the meta-analysis in this study were performed using R programming software.” What do the authors mean by “all statistical analysis involving the meta-analysis”? All analyses should be related to this meta-analysis or not?
We have rewritten as follows: “All meta-analyses in this study were performed using R programming software” (Page 3, line 144 in the version without track changes; Page 4, line 169 in the version with track changes) under the meta-analysis section in the methods section.
- The authors repeated several times the following sentence: “The meta-analysis was performed for the liver parameters (IHL, ALT, AST), plasma lipid profile parameters (HDL-C, LDL-C, TC, TG) and glucose metabolism parameters (fasting glucose, fasting insulin, HbAC1).” Please try to avoid repetition.
Thank you for pointing this out. We have rephrased the sentences as follows to avoid the repetition:
”Cumulative meta-analyses were used to explore the effects of exercise on NAFLD-related liver parameters (IHL, ALT, and AST), plasma lipid profile parameters (HDL-C, LDL-C, TC, and TG), and glucose metabolism parameters (fasting glucose, fasting insulin, and HbA1c).” (Page 3, line 145 to 148 in the version without track changes; Page 4, line 170 to 174 in the version with track changes).
- It seems that all studies included in the present meta-analysis measured IHL and the liver enzyme ALT (as stated by the authors). Then, the authors stated the following: “…the liver enzyme AST was not analyzed in two studies [22,43]. HDL-C and LDL-C analysis were performed by five studies [41-45]. Furthermore, TG was not mentioned in one study [39] and TC in two studies [39,40]. Fasting glucose and insulin concentrations were not measured by two studies [39,43]. HbA1c analysis was conducted by five studies [22,38,40,45,46].”
So, can any conclusion be drawn? Please consider that some parameters were available only in 2 studies. This is a very important limitation that should be mentioned among the limitations.
Thank you for your comment. We have now added it in the limitations as follows: “Out of the 10 included studies, not every study measured all the different NAFLD-related parameters considered in this study and in some cases, some parameters were only available in two studies. Moreover, only two of the included studies conducted a combination of aerobic and resistance training. Therefore, a conclusion based on the me-ta-analysis in these subgroups with few studies might not reflect the actual effect.” (Page 25, lines 662 to 667 in the version without track changes; Page 28, lines 842 to 847 in the version with track changes)
Yet, the authors stated later: “Data on IHL was available from all studies. However, only those studies that reported IHL as percentage values (n=8) were considered for meta-analysis.” Please be careful and try to be precise as much as possible. There are several inconsistencies and it seems that the first statement (that 10 studies with 316 individuals were included), actually is not true. Please correct this accordingly.
This has been clarified in the result section (3.2.1). The sentences have been re-written as follows: “All the studies that measured IHL using MRS (n=8) were included for the meta-analysis. Two other studies were not considered as they used US, which is not comparable with the results of MRS [32,38].” (Page 10, lines 251 to 253 in the version without track changes; Page 10, lines 306 to 311 in the version with track changes)
8.“Effect of all the exercise interventions on body weight change and fat mass was non-significant for all the studies. Therefore, significant changes in the NAFLD-related clinical parameters in exercise groups compared with control reported in our study were all independent of weight loss and fat mass reduction.” Did the authors consider fat mass reduction too? It has not been mentioned. I did not find any results reported on body weight or fat mass. Do the authors think that such changes can be correlated with changes in other parameters (e.g. lipids)?
We have now analysed different anthropometry changes after the exercise intervention including body weight, whole body fat mass, body fat percentage, lean body mass, BMI, visceral adipose tissue, and waist circumference. Since these changes were non-significant (unpaired two-tailed t-test; p>0.05), except for one study that had significant reduction waist circumference (Pugh et al., 2014), we concluded that all changes in the NAFLD-related clinical parameters in exercise groups compared with control reported in our study were all independent of anthropometry changes.
We have included these analyses in results section 3.1 as follows: “Anthropometry changes after the exercise intervention were analyzed using an unpaired two-tailed t-test. Of all the anthropometry changes reported in the studies, the body weight, whole body fat mass, body fat percentage, lean body mass, BMI, and visceral adipose tissue were all non-significantly changed after the different exercise interventions (p > 0.05). The results from only one study [36] showed that the aerobic exercise intervention significantly reduced the waist circumference (p=0.01), while the others remained non-significantly changed (p>0.05).” (Page 5, lines 204 to 210 in the version without track changes; Page 5, lines 240 to 246 in the version with track changes)
- The authors stated: “For IHL, only studies using MRS as detecting hepatic fat were included in the meta-analysis.” Please clarify this point and explain why did you decide this, when previously you stated that IHL was available in all studies?
We have now clarified this point as follows: “All the studies that measured IHL using MRS (n=8) were included for the meta-analysis. Two other studies were not considered as they used US, which is not comparable with the results of MRS [32,38].” (Page 10, lines 251 to 253 in the version without track changes; Page 10, lines 306 to 311 in the version with track changes).
We hope it is now clearer.
9.I would suggest to present more clearly Forest plots in the results section, they are not readable. Overall, results section can be presented more clearly and shortly.
We have modified the plots to make them more readable by increasing their size.
10.One of the most common statement in the results section is: “Five, one, and two studies focused on aerobic exercise, resistance training, and a combination of these, respectively.” This point should be clearly stated in the material and methods section. Considering different types of exercises (aerobic exercise, resistance training, and their combination) how many studies are there?
This is now clearer in the method section. We have added a table (Table 2) to avoid repeating the statement in the text.
Table 2. Included studies for meta-analysis for each clinical parameter
|
Clinical Parameter |
Aerobic Exercise |
Resistance Training |
Exercise combination (Aerobic exercise + Resistance training) |
Overall exercise (Aerobic exercise + Resistance training + Exercise combination) |
|
IHL |
5 |
1* |
2 |
8 |
|
Liver stiffness |
- |
- |
1* |
1* |
|
ALT |
6 |
2 |
2 |
10 |
|
AST |
5 |
2 |
2 |
9 |
|
HDL-C |
3 |
1* |
1* |
5 |
|
LDL-C |
2 |
1* |
1* |
4 |
|
TC |
3 |
2 |
2 |
7 |
|
TG |
5 |
2 |
2 |
9 |
|
Fasting glucose |
4 |
2 |
2 |
8 |
|
Fasting insulin |
4 |
2 |
2 |
8 |
|
HbA1c |
2 |
2 |
1* |
5 |
|
IHL: intrahepatic lipid content, ALT: alanine aminotransaminase, AST: aspartate aminotransaminase, HDL-C: high-density lipoprotein cholesterol, LDL-C: low-density lipoprotein cholesterol, TC: total cholesterol, TG: triglyceride, HbA1c: glycated hemoglobin. Overall exercise includes all the studies which measured the clinical parameter* No meta-analysis could be performed |
||||
(Page 9, line 247 in the version without track changes; Pages 9 to 10, line 294 in the version with track changes)
- The authors stated: “Each intervention arm was compared with the control arm and divided into its subgroups (aerobic exercise alone - 6 RCTs, resistance training alone – 3RCTs, and a combination of aerobic exercise and resistance training– 2 RCTs).” Does this mean that “exercise overall” means all these subgroups together? I suggest to define the meaning of “exercise” or “exercise overall” in order to better understand what the authors discussed later about the changes after exercise (e.g. “Heterogeneity in the effect of exercise on ALT was high” (which one? overall exercise?)).
The definition of overall exercise has been included in the method section under point 2.5.2. Meta-analysis. We have stated that “The results for each of the parameters were represented as forest plots for overall exercise, along with subgroup analyses. Overall exercise included all the studies that measured the clinical parameter. The subgroup analyses were based on the exercise regimen (aerobic exercise/resistance training/combination of aerobic exercise and resistance training).” (Page 4, lines 156 to 159 in the version without track changes; Page 4, lines 182 to 187 in the version with track changes).
For a better understanding a new table (Table 2) has been inserted as well.
Table 2. Included studies for meta-analysis for each clinical parameter
|
Clinical Parameter |
Aerobic Exercise |
Resistance Training |
Exercise combination (Aerobic exercise + Resistance training) |
Overall exercise (Aerobic exercise + Resistance training + Exercise combination) |
|
IHL |
5 |
1* |
2 |
8 |
|
Liver stiffness |
- |
- |
1* |
1* |
|
ALT |
6 |
2 |
2 |
10 |
|
AST |
5 |
2 |
2 |
9 |
|
HDL-C |
3 |
1* |
1* |
5 |
|
LDL-C |
2 |
1* |
1* |
4 |
|
TC |
3 |
2 |
2 |
7 |
|
TG |
5 |
2 |
2 |
9 |
|
Fasting glucose |
4 |
2 |
2 |
8 |
|
Fasting insulin |
4 |
2 |
2 |
8 |
|
HbA1c |
2 |
2 |
1* |
5 |
|
IHL: intrahepatic lipid content, ALT: alanine aminotransaminase, AST: aspartate aminotransaminase, HDL-C: high-density lipoprotein cholesterol, LDL-C: low-density lipoprotein cholesterol, TC: total cholesterol, TG: triglyceride, HbA1c: glycated hemoglobin. Overall exercise includes all the studies which measured the clinical parameter* No meta-analysis could be performed |
||||
(Page 9, line 247 in the version without track changes; Pages 9 to 10, line 294 in the version with track changes)
We have also used the “overall exercise” terminology while describing about the heterogeneity in the effect of exercise on the different clinical parameters.
This point is very important as the aim was to distinguish the contribution of different exercise regimens to the different clinical parameters.
Additionally, did each arm has its own control? What were healthy status and lifestyle of the controls?
Thank you for asking these questions. Yes, each arm has a control arm, because we included only randomized controlled trials (RCTs). For one study (Shamsoddini et al.) each of the intervention arm was compared with the control arm as stated in the results under point 3.1. research results.
“One out of the 10 studies had two exercise intervention arms with different exercise regimes—aerobic exercise and resistance training [32]—which were counted as separate RCT exercise intervention arms. Each intervention arm was compared with the control arm and divided into its subgroups (aerobic exercise alone—6 RCTs; resistance training alone—3RCTs; and a combination of aerobic exercise and resistance training—2 RCTs).”
(Page 4, lines 189 to 194 in the version without track changes; Page 5, lines 222 to 227 in the version with track changes):
The control group NAFLD diagnosed subjects, who had no diet modification changes. This has been mentioned in the Inclusion and exclusion criteria section 2.3 as follows:
“Articles were included if they met the following inclusion criteria: (1) RCTs in the adult population (age ≥ 18years) written in English; (2) study subjects diagnosed with NAFLD through biopsy, US, magnetic resonance imaging /magnetic resonance spectroscopy (MRS) or computed tomography; (3) studies with any sort of supervised physical activity for at least four weeks; (4) no diet modification in both exercise intervention and control groups. “(Page 3, lines 102 to 107 in the version without track changes; Page 3, lines 122 to 127 in the version with track changes)
11.Similarly, the authors stated: “The studies included in the analysis enrolled between 6 and 33 participants in each arm. The duration of the studies was between 8-16 weeks.” This should be present in details. The duration of exercise intervention is not clear (between 8 to 16 weeks, please report the mean). In addition, given that all parameters were not present in all studies included (10), it is so difficult to decide the mean duration, eventually it should be mentioned. Then, how could you compare 6 vs 33 participants? Pleas clarify.
Thank you for your comment.
As correctly indicated by the reviewer that all parameters were not present in all included studies and that a mean duration would be difficult to determine, we have now included the median duration of the exercise intervention and the median of sample sizes. This measure could possibly rule out any bias contributed by extreme values.
We understand that there is diversity in the number of study participants. Hence, to have a meaningful comparison, we used the standardized mean difference approach. Here, the standard deviations (SDs) are together with the sample sizes to compute the weight given to each study. Studies with small SDs are given relatively higher weight whilst studies with larger SDs are given relatively smaller weight.
- In discussion section please discuss your findings in more details. For instance, the authors stated: “These results were also confirmed by several other systematic reviews with different included studies”. Please discuss briefly what were the different inclusion criteria? What was different and what was similar to your study?
We have discussed our findings in more detail. We have modified the text as follows: “In addition to our study, many other SRs, which also included non-supervised studies, shorter resistance training sessions, longer aerobic interventions, and higher intensity aerobic interventions, reported similar IHL reductions [16,45-48].” (Page 23, lines 522 to 525 in the version without track changes; Page 24, lines 604 to 609 in the version with track changes)
In general, compare more your findings with others’ findings instead of just mention it (e.g. contradictory results were obtained by Keating’s meta-analysis [53]. They found out that IHL, but not ALT, was reduced after overall exercises. The reason for such contradictory and inconsistent results may be attributed to the differences in the included studies, exercise regime, intensity, duration, and frequency). Please summarize what were the main differences in selection of the studies, exercise regime etc., and support your own findings. Overall, discussion should be shorter and more readable.
Thank you for pointing this out. We have modified the discussion section and discussed our findings in more detail.
We have modified the text as follows: “Contradictory results were obtained by Keating’s meta-analysis, where shorter resistance training sessions were considered [58]. They found that IHL, but not ALT, was reduced after overall exercise. The limited inclusion of studies on aerobic exercise by Keatings et al. might have contributed to the contradictory results.” (Page 23, lines 554 to 558 in the version without track changes; Page 25, lines 677 to 681 in the version with track changes)
- The authors stated: “The significant reductions in TG concentrations in the resistance training subgroup could be partly explained by exercised-induced myokine production such as irisin, which is negatively correlated with TG, TC, and intra hepatic triglycerides [90,91].” The authors did not study exercised-induced myokine production such as irisin, so I am not sure this is appropriate to mention in the present form.
We totally agree with this comment and we have adjusted the sentence as follows “The significant reductions in TG concentrations in the resistance training subgroup could be partly explained by exercise-induced myokine production such as irisin, which was negatively correlated with TG, TC, and intrahepatic triglycerides [68,69]. Irisin increases total energy expenditure and modulates lipid metabolism by inhibiting enzymes such as sterol regulatory element-binding protein-1c and fatty acid synthases in hepatocytes [70]. Irisin decreases in NAFLD patients [68] but can increase in obese people after resistance training, though not with aerobic training [71]. This mechanism could plausibly explain the increase in TG in the resistance training group. However, as irisin was not measured in any of the included studies, a further validation was not possible.” (Page 24, lines 584 to in 593 version without track changes; Page 26, lines 720 to 733 in the version with track changes)”.
- This is one more example of the repetition through the manuscript: “Only IHL content quantified by MRS was used for performing the statistical analyses. Therefore, only 8/10 studies were included in the meta-analysis.” There are several repetitions, please check it carefully.
We agree with this comment. We have deleted this sentence and pointed it out more in detail in the results. We have mentioned as follows: “All the studies that measured IHL using MRS (n=8) were included for the meta-analysis. Two other studies were not considered as they used US, which is not comparable with the results of MRS [32,38].” (Page 10, lines 251 to 253 in the version without track changes; Page 10, lines 306 to 311 in the version with track changes)
- Given that this meta-analysis did not include data on gut microbiota, there is no need to discuss widely about that. I suggest to make just a brief discussion what is missing in the literature and what is expecting to be reported in the field in the following years. Consequently, limits of the present study should be presented in more details.
Thank you for your comment. We agree that this meta-analysis did not include any studies related to the gut microbiota. However, we would still like to mention it because it was one of the concerned outcomes of interest in our study. Nevertheless, we have shortened the discussion related to this topic and focused more on comparing our results with other studies.
- Do not the authors believe that the conclusions are general and limited on only few studies included? Considering that all parameters were not present in all studies, a final conclusion cannot be drawn (as also has been mentioned by the authors). Please try to reword this section.
Thank you for this comment. We have reworded the conclusion as follows: “Our meta-analysis concludes that exercise overall has a beneficial effect on alleviating NAFLD without significant weight loss. After overall exercise, IHL, ALT, AST, LDL-C and TG were reduced. Aerobic exercise seemed to alleviate NAFLD-related liver parameters (IHL, ALT and AST), while resistance training was more effective at alleviating TG and TC. The combination of aerobic and resistance training alleviated only IHL. These results could allow the establishment of exercise (or lifestyle modification) guidelines where weight loss is not the focus but could be a consequence of the lifestyle modification in some patients. It must be noted, however, that our conclusions are based on only a small number of studies with small sample sizes. Therefore, more studies are warranted for a comprehensive and meaningful comparison of different exercise regimes, the long-term effects of exercise, and the gut microbiota and metabolites’ contribution.” (Page 26, lines 676 to 686 in the version without track changes; Page 29, lines 856 to 875 in the version with track changes)
19.Please check carefully English language as well as the abbreviations through the text.
We have checked the abbreviations. We also had our manuscript professionally edited by MDPI’s English editing service.
Reviewer 2 Report
This paper reviews the literature and attempts a meta- analysis on the effect of exercise (in the absence of dietary modification) upon a variety of tests of liver and carbohydrate metabolism and cardiovascular risk. It also attempts to tease out differences between the effects of aerobic and resistance exercise on these clinically-relevant metabolic parameters.
This is a very important area of clinical practice where many patients, keen to improve their health, increase their exercise levels without achieving weight loss and the metabolic benefits of that exercise are unclear.
The authors' approach is rigorous and well-described. The analysis is also rigorous. The use of English is poor especially their use of grammar (inappropriate use and omission of definite and indefinite articles, inappropriate matching of singular subjects with plural verbs).
The strengths and limitations are fairly described (713-733) but I would add to the limitations in that:
1) Dietary fat content changes liver fat content. "No diet modifications" was an inclusion criterion (116). Food diaries were collected for many of the subjects (Table 1; 222) Could the authors please comment upon diet and the measured outcomes (more than they already do on line 725). Why was "no dietary modification" stipulated and what consequences do they believe arise from choosing studies as they did? Did any significant dietary changes occur on the basis of those diaries even though unintended?
2) Anthropometry changes liver fat content - especially visceral fat mass. Some subjects were recruited because they were obese; I assume others were in the healthy weight range. No clear mention is made of anthropometry changes with exercise aside from weight. Are there anthropometric measures common to some or all papers allowing us to determine changes in body composition with exercise? There is an indirect reference to a lack of change in fat mass (243) and I assume this was whole body fat mass(?). How was fat mass measured? I assume, even if non-significant, muscle mass increased with supervised exercise and therefore fat mass decreased (if body weight remained constant). Some more data and some discussion on this point would be welcome even if the anthropometric data (e.g. waist circumference?) are unavailable. It would help the reader to understand what did change in body composition terms (apart from IHL) if weight did not change.
3) I doubt the literature is of sufficient quality and quantity to date to allow meaningful conclusions to be made when comparing the different types of exercise on these metabolic parameters. Any exercise is good even if no weight loss is achieved.
Minor point:
4) The mathematics is not quite right: (lines 194-196) 73-63=12. Also in Figure 1 38-18=10.
Author Response
lly relevant studies. After removing duplicates and screening through article titles and abstracts, 73 studies were retained for full-text reading. After excluding 61 ineligible studies based on the exclusion/inclusion criteria, 12 RCTs remained for review. Two more studies were excluded after no response from the authors for raw data to be included in the meta-analysis.” (Page 4, lines 183 to 187 in version without track changes; Page 5, lines 215 to 220 in version with track changes)
Dear Reviewer, thank you for the helpful comments. We have tried our best to revise the manuscript according to your suggestions and comments:
The strengths and limitations are fairly described (713-733) but I would add to the limitations in that:
- Dietary fat content changes liver fat content. "No diet modifications" was an inclusion criterion (116). Food diaries were collected for many of the subjects (Table 1; 222) Could the authors please comment upon diet and the measured outcomes (more than they already do on line 725). Why was "no dietary modification" stipulated and what consequences do they believe arise from choosing studies as they did? Did any significant dietary changes occur on the basis of those diaries even though unintended?
Thank you for the question. Diet plays an important role in NAFLD development. Lifestyle modifications focusing on increased physical activity along with dietary modifications are currently the treatment according to guidelines for NAFLD individuals. In our manuscript, we aimed to see the exclusive benefits that an exercise intervention could offer in alleviating NAFLD. Therefore, we excluded any additional diet intervention that could possibly cause changes in liver, lipid and glucose parameters.
As correctly pointed out, several studies reported the use of food diaries analysed by a dietitian. On the basis of these food diaries, no occurrences of significant dietary changes were reported. We have reported as follows “Some studies had food records, but no diet results were reported [34-38]. All studies stated no diet changes.” (Page 4, lines 200 to 201 in version without track changes; Page 5, lines 235 to 237 in version with track changes)
Anthropometry changes liver fat content - especially visceral fat mass. Some subjects were recruited because they were obese; I assume others were in the healthy weight range. No clear mention is made of anthropometry changes with exercise aside from weight. Are there anthropometric measures common to some or all papers allowing us to determine changes in body composition with exercise? There is an indirect reference to a lack of change in fat mass (243) and I assume this was whole body fat mass(?). How was fat mass measured? I assume, even if non-significant, muscle mass increased with supervised exercise and therefore fat mass decreased (if body weight remained constant). Some more data and some discussion on this point would be welcome even if the anthropometric data (e.g. waist circumference?) are unavailable. It would help the reader to understand what did change in body composition terms (apart from IHL) if weight did not change.
Thank you for your comments. The inclusion criteria for the study participant was NAFLD diagnosis while obesity was not a necessary inclusion criterion. To help the readers understand the change in body composition, we have n included the analysis of different anthropometry changes after the exercise intervention commonly reported in the studies including body weight, whole body fat mass, body fat percentage, lean body mass, BMI, visceral adipose tissue, and waist circumference. The fat mass reported in our study was referring to the whole-body fat mass which is usually measured using air displacement plethysmography, body composition analyser or dual energy x-ray absorptiometry. Since these changes were mainly non-significant (unpaired two-tailed t-test; p>0.05), we concluded that all changes in the NAFLD-related clinical parameters in exercise groups compared with control reported in our study were all independent of anthropometric changes.
We have included these analyses in the results section 3.1 as follows: “Anthropometry changes after the exercise intervention were analysed using an unpaired two-tailed t-test. Of all the anthropometry changes reported in the studies, the body weight, whole body fat mass, body fat percentage, lean body mass, BMI, and visceral adipose tissue were all non-significantly changed after the different exercise interventions (p > 0.05). The results from only one study [36] showed that the aerobic exercise intervention significantly reduced the waist circumference (p=0.01), while the others remained non-significantly changed (p>0.05).” (Page 5, lines 204 to 210 in version without track changes; Page 5, lines 240 to 246 in version with track changes)
- I doubt the literature is of sufficient quality and quantity to date to allow meaningful conclusions to be made when comparing the different types of exercise on these metabolic parameters. Any exercise is good even if no weight loss is achieved.
We agree on that and re-phrased the conclusion as follows: “Our meta-analysis concludes that exercise overall has a beneficial effect on alleviating NAFLD without significant weight loss. After overall exercise, IHL, ALT, AST, LDL-C and TG were reduced. Aerobic exercise seemed to alleviate NAFLD-related liver parameters (IHL, ALT and AST), while resistance training was more effective at alleviating TG and TC. The combination of aerobic and resistance training alleviated only IHL. These results could allow the establishment of exercise (or lifestyle modification) guidelines where weight loss is not the focus but could be a consequence of the lifestyle modification in some patients. It must be noted, however, that our conclusions are based on only a small number of studies with small sample sizes. Therefore, more studies are warranted for a comprehensive and meaningful comparison of different exercise regimes, the long-term effects of exercise, and the gut microbiota and metabolites’ contribution.” (Page 26, lines 676 to 686 in version without track changes; Page 29, lines 856 to 875 in version with track changes)
Minor point:
- The mathematics is not quite right: (lines 194-196) 73-63=12. Also in Figure 1 38-18=10.
Thank you for pointing this out. The numbers in the Figure 1 (Page 5, line 212 in version without track changes; Page 6, line 248 in version with track changes) and the text have been corrected as follows “The electronic search (until February 13, 2020) yielded 1949 potentially relevant studies. After removing duplicates and screening through article titles and abstracts, 73 studies were retained for full-text reading. After excluding 61 ineligible studies based on the exclusion/inclusion criteria, 12 RCTs remained for review. Two more studies were excluded after no response from the authors for raw data to be included in the meta-analysis.” (Page 4, lines 183 to 187 in version without track changes; Page 5, lines 215 to 220 in version with track changes)
We also had our manuscript professionally edited by MDPI’s English editing service.
Round 2
Reviewer 2 Report
Concerns adequately addressed.